# Large-scale genomic analyses reveal insights into pleiotropy across circulatory system diseases and nervous system disorders

Xinyuan Zhang[1,2], Anastasia M. Lucas[1], Yogasudha Veturi[1], Theodore G. Drivas [1], William P. Bone[1,2], Anurag Verma[1], Wendy K. Chung[3], David Crosslin[4], Joshua C. Denny[5,6], Scott Hebbring[7], Gail P. Jarvik[4], Iftikhar Kullo [8], Eric B. Larson[9], Laura J. Rasmussen-Torvik[10], Daniel J. Schaid[11], Jordan W. Smoller [12], Ian B. Stanaway [4], Wei-Qi Wei[6], Chunhua Weng [13] & Marylyn D. Ritchie [1✉]

Clinical and epidemiological studies have shown that circulatory system diseases and nervous system disorders often co-occur in patients. However, genetic susceptibility factors shared between these disease categories remain largely unknown. Here, we characterized pleiotropy across 107 circulatory system and 40 nervous system traits using an ensemble of methods in the eMERGE Network and UK Biobank. Using a formal test of pleiotropy, five genomic loci demonstrated statistically significant evidence of pleiotropy. We observed region-specific patterns of direction of genetic effects for the two disease categories, suggesting potential antagonistic and synergistic pleiotropy. Our findings provide insights into the relationship between circulatory system diseases and nervous system disorders which can provide context for future prevention and treatment strategies.

[1] Department of Genetics and Institute for Biomedical Informatics, Perelman School of Medicine, University of Pennsylvania, Philadelphia, PA 19104, USA. [2] Genomics and Computational Biology Graduate Group, Perelman School of Medicine, University of Pennsylvania, Philadelphia, PA 19104, USA. [3] Department of Pediatrics and Medicine, Columbia University, New York, NY 10032, USA. [4] Department of Biomedical Informatics and Medical Education, University of Washington, Seattle, WA 98109, USA. [5] Department of Medicine, Vanderbilt University, Nashville, TN 37235, USA. [6] Department of Biomedical Informatics, Vanderbilt University, Nashville, TN 37230, USA. [7] Center for Human Genetics, Marshfield Clinic, Marshfield, WI 54449, USA. [8] Division of Cardiovascular Diseases, Mayo Clinic, Rochester, MN 55905, USA. [9] Kaiser Permanente Washington Health Research Institute, Seattle, WA 98101, USA. [10] Department of Preventive Medicine, Northwestern University Feinberg School of Medicine, Chicago, IL 60611, USA. [11] Division of Biomedical Statistics and Informatics, Department of Health Sciences Research, Mayo Clinic, Rochester, MN 55905, USA. [12] Psychiatric and Neurodevelopmental Genetics Unit, Massachusetts General Hospital, Boston, MA 02114, USA. [13] Department of Biomedical Informatics, Columbia University, New York, NY 10032, USA. ✉email: marylyn@pennmedicine.upenn.edu

Circulatory system diseases and nervous system disorders have a significant impact on mortality worldwide. Because of the distinct disease manifestations, diseases in these categories have long been diagnosed, treated, and studied independently. However, for decades, clinicians and researchers have noted a link between circulatory system diseases and nervous system disorders. For instance, it is clear that cardiac pathologies can be produced as a result of neurological illness[1]. Heart failure is a potential risk factor for Alzheimer's disease[2] and occurs more than twice as often in Parkinson's disease patients compared to non-Parkinson's disease patients[3]. However, the genetic variants influencing both disease categories are largely unknown.

One of the potential genetic links can be via pleiotropy, a phenomenon by which a gene or a genetic variant influences more than one phenotypic trait[4]. Pleiotropy has long been recognized in model organisms[5], and its ubiquitous role has recently been appreciated in the human genome—90% of genome-wide association study (GWAS) loci are pleiotropic[6,7]. The definition of pleiotropy in this manuscript refers to 'statistical pleiotropy.' which describes a genetic variant that is statistically associated with more than one phenotype/trait[6]. Large-scale biobanks, coupled with Electronic Health Records (EHRs), offer unprecedented opportunities to study pleiotropy. Nevertheless, most studies of pleiotropy in the biomedical literature thus far are solely inferred from GWAS studies[6–9] in multiple independent datasets. For instance, a global overview of pleiotropy across phenotypes with high disease prevalence has been demonstrated using GWAS summary statistics[6], highlighting the extent of potential pleiotropy across broad disease categories. However, genetic variants that contribute to a wide spectrum of diseases (including the less common ones) across specific disease categories have not been extensively studied.

Methods for detecting pleiotropy can be broadly grouped into univariate and multivariate categories. Univariate methods test the association between one genetic variant and one phenotype per statistical model. Phenome-wide association studies (PheWAS) are among the most commonly used univariate methods that examine the impact of genetic variants across a broad range of phenotypes using univariate regression models[10]. The application of PheWAS has uncovered novel potential pleiotropy using EHR phenotypes in many prior studies[11–15]. Additional univariate methods in the literature also refer to a combined analysis of summary statistics obtained from multiple GWAS studies[16–22]. Multivariate methods, or multi-trait joint methods, refer to the inclusion of two or more phenotypes in the association test in the same statistical model[4]. Multivariate methods have demonstrated increased power for detecting pleiotropy but have not been widely applied on large-scale natural biomedical datasets. In this study, we used MultiPhen[23] as our multi-trait joint analysis method as it is designed for binary phenotypes and has shown sufficient statistical power[24]. MultiPhen analyzes multiple phenotypes simultaneously by testing the linear combination of phenotypes with the genotype using an ordinal regression model. In general, multivariate methods are more powerful than combining univariate GWAS summary statistics[25]. Since no single method can detect all types of genotype-phenotype relationships in natural biomedical data, it has been suggested to apply both univariate and multivariate methods[25] and to view them as complementary approaches[26]. This is the strategy we adopted in our study design.

In this study, we aimed to characterize pleiotropy specifically across circulatory system diseases and nervous system disorders. We have applied genome-wide PheWAS and MultiPhen analyses on 43,015 European-ancestry adults from the eMERGE network, followed by a systematic replication analysis in 295,423 European-ancestry participants from the UK Biobank (UKBB) (Supplementary Fig. 1). This effort yielded a comprehensive comparison of the characteristics of applying univariate and multivariate methods on independent biobank datasets. To investigate pleiotropy, we further performed a formal statistical test of pleiotropy, which pinpoints precisely which specific phenotypes show evidence of pleiotropy via performing multivariate analyses iteratively using a method called Pleio[27]. Through these analyses, we have provided evidence to explain the relationship between circulatory system diseases and nervous system disorders that can be characterized as pleiotropic, recognizing that we observed both synergistic and antagonistic pleiotropy between these disease categories.

## Results

**Phenotypic characterization.** The eMERGE Phase III dataset consists of 99,185 subjects coupled with longitudinal EHR data from the United States. The UKBB has genotypic and phenotypic data on 487,409 individuals from the United Kingdom. Our phenotypes of interest are a comprehensive set of circulatory system diseases and nervous system disorders.

The phenotypes are defined by utilizing the International Classification of Diseases (ICD) diagnosis codes obtained from the EHR. Because of the differences in disease coding practices and regulations between the US and the UK, the composition of ICD codes differs between the two datasets. The eMERGE network has mostly (~82%) ICD-9-CM codes, while the UKBB has predominantly (~98%) ICD-10 codes. However, to our current knowledge, there is no available official equivalence mapping that maps ICD codes between the UK and the US, given that the US uses its own national variation of ICD codes (known as ICD-CM). To address this for our replication study design, we collected the ICD codes from the official website in three broad categories: 'mental disorders', 'disease of the nervous system', and 'disease of circulatory system', used the disease categories provided by ICD to assign the ICD-9-CM and ICD-10 codes into their respective categories, and then manually curated a common list of phenotypes that are present in both eMERGE and UKBB.

We excluded phenotypes based on the following criteria: 1. Disease that was secondary to environmental or comorbid causes such as drug or injury; 2. Childhood-onset developmental and psychiatric disorders; and 3. Diseases mainly occurring in organs other than heart and brain (such as the limbs). We applied a minimum case number threshold of 200 to ensure adequate statistical power of the association tests[28]. In this study, we use the term "nervous system disorders" to refer to mental disorders and diseases of the nervous system[29]. In total, we curated 40 and 25 nervous system diseases in eMERGE and UKBB, respectively; 107 and 77 circulatory system diseases in eMERGE and UKBB, respectively (Supplementary Data 1). These phenotypes are categorized into seven groups of circulatory system diseases and seven groups of nervous system disorders (Supplementary Data 1).

**Discovery and replication of univariate and multivariate associations.** After quality control, genome-wide PheWAS and MultiPhen analyses were performed on 43,015 European-ancestry adults and 7,629,801 common SNPs across 147 phenotypes in the eMERGE network. A formal systematic replication analyses was conducted in UKBB on 134,363 genetic variants that had an exploratory $p$-value significance of $p \leq 1 \times 10^{-4}$ from analyses in eMERGE dataset (and passed QC in the UKBB dataset). The use of an exploratory $p$-value threshold such as $1 \times 10^{-4}$ enables exploration of genetic variants beyond the most significant signals at a genome-wide significance threshold. Other studies have employed this strategy and it can be beneficial to identify variants

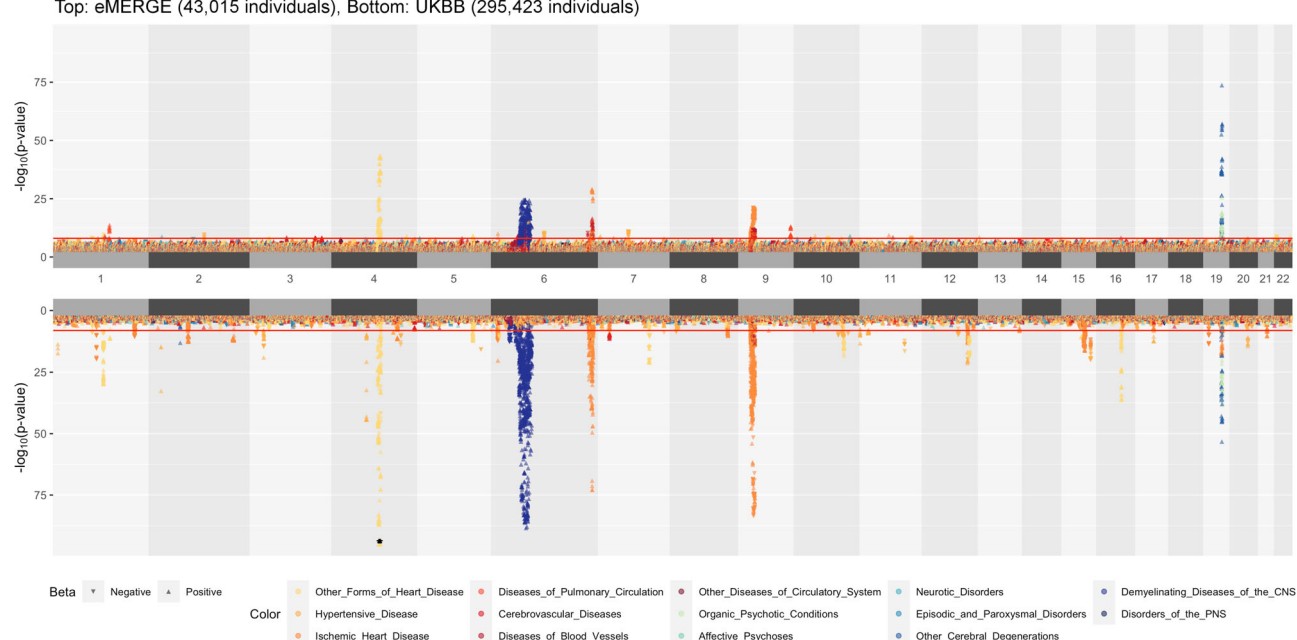

**Fig. 1 Landscape of PheWAS results.** A position-to-position comparison of PheWAS results between eMERGE and UKBB. eMERGE PheWAS was performed genome-wide as the discovery analysis. UKBB PheWAS included the SNPs with $p \leq 1 \times 10^{-4}$ across all tested phenotypes in eMERGE; only these SNPs were evaluated in the replication analysis. The direction of each triangle indicates the direction of genetic effect. Colors denote various disease groups. The assignment of ICD codes to disease groups can be found in Supplementary Data 1. The red line indicates the genome-wide significance threshold $p$-value of $1 \times 10^{-8}$. To reduce the margin induced by the extremely small $p$-values, we have collapsed SNPs with $p$-value $< 1 \times 10^{-95}$ into one overlapping triangle indicated by an asterisk on chromosome 4 for UKBB.

that may not meet genome-wide significance in one dataset but otherwise be potentially informative[12].

From PheWAS results for eMERGE and UKBB (Fig. 1), we found that the top association signals from eMERGE analyses are reproducible in the UKBB replication dataset, many of which serve as positive controls as they were discovered in previous studies in the literature. For instance, we observed that SNPs located on chromosome 4q25 are significantly associated with atrial fibrillation in eMERGE and replicated in UKBB. In particular, we replicated a previously reported SNP rs2200733 near *PIXT2* gene (eMERGE $p$-value: $5.898 \times 10^{-37}$, UKBB $p$-value: $7.112 \times 10^{-142}$) that was shown to be significantly associated with atrial fibrillation among individuals of European-ancestry[30]. We also identified SNPs near the *APOE* gene at 19q13.32 to be associated with Alzheimer's disease and dementia; of these, we replicated a previously reported SNP, rs429358, as our most statistically significant SNP (discovery eMERGE $p$-value: $1.604 \times 10^{-74}$, replication UKBB $p$-value: $6.327 \times 10^{-54}$) associated with Alzheimer's disease[31]. Similarly, we found a previously-detected association between SNP rs1333049 near *CDKN2B-AS1* (discovery eMERGE $p$-value: $6.016 \times 10^{-22}$, replication UKBB $p$-value: $7.982 \times 10^{-77}$) and coronary artery disease[32], and found SNPs in the *HLA* region to be highly associated with multiple sclerosis[33].

In the UKBB replication dataset, we observed lower $p$-values (higher significance levels) for many genetic regions that showed moderate significance ($1 \times 10^{-8} \leq p$-value $\leq 0.001$) in the eMERGE dataset. For example, SNPs on chromosome 4 that were moderately associated with essential hypertension in the eMERGE network demonstrated a very strong statistical significance of association in the UKBB. Similar noticeable association signals were observed in UKBB across the genome (Fig. 1). Overall, the UKBB PheWAS identified 286 loci (Fig. 2: $218 + 57 + 11 = 286$) from the discovery eMERGE PheWAS (out

of 35,352 SNPs that were evaluated in the UKBB replication PheWAS) using an exploratory $p$-value threshold (Fig. 2).

The landscape of MultiPhen results is shown in Fig. 3. Most of the strong association signals that were observed in PheWAS (Fig. 1) were also significant in MultiPhen analyses. As with the PheWAS results, MultiPhen identified previously known SNPs in both datasets, including the previously-mentioned rs2200733 (eMERGE multi-trait joint $p$-value: $8.305 \times 10^{-16}$, UKBB multi-trait joint $p$-value: $5.873 \times 10^{-82}$), rs429358 (eMERGE multi-trait joint $p$-value: $3.137 \times 10^{-48}$, UKBB multi-trait joint $p$-value: $3.888 \times 10^{-49}$) and rs1333049 (eMERGE multi-trait joint $p$-value: $1.309 \times 10^{-15}$, UKBB multi-trait joint $p$-value: $6.208 \times 10^{-62}$). Compared to PheWAS results, there is a lower number of significant loci identified by MultiPhen, especially in the eMERGE analysis (Fig. 2). One of the reasons for this observation is that the univariate method is slightly more powerful than a multivariate method when the genetic effects are consistent with the phenotypic correlation[23], and the inclusion of uncorrelated phenotypes in a model may reduce the association signal for MultiPhen. To extract how many unique SNPs were significant in the discovery and replication analyses using univariate (PheWAS) and multivariate (MultiPhen) approaches, we created an UpSet[34] plot (Fig. 2). For example, in eMERGE, 40 loci passed the exploratory $p$-value threshold ($1 \times 10^{-4}$) in both PheWAS and MultiPhen analyses (Fig. 2: $29 + 11 = 40$). For UKBB, there were 68 loci that passed the $p$-value threshold ($1 \times 10^{-4}$) in both PheWAS and MultiPhen results (Fig. 2: $57 + 11 = 68$).

We characterized the 11 loci (607 SNPs) that had statistically significant associations with at least one phenotype in both eMERGE and UKBB via *both* PheWAS and MultiPhen (Fig. 2 – column 5 of UpSet plot). These SNPs mapped to 32 genes using the RefSeq database[35] in ANNOVAR[36] (Supplementary Data 2 and Supplementary Fig. 2A). A total of 2 loci (76 SNPs) met a Bonferroni correction for multiple testing burden (Supplementary

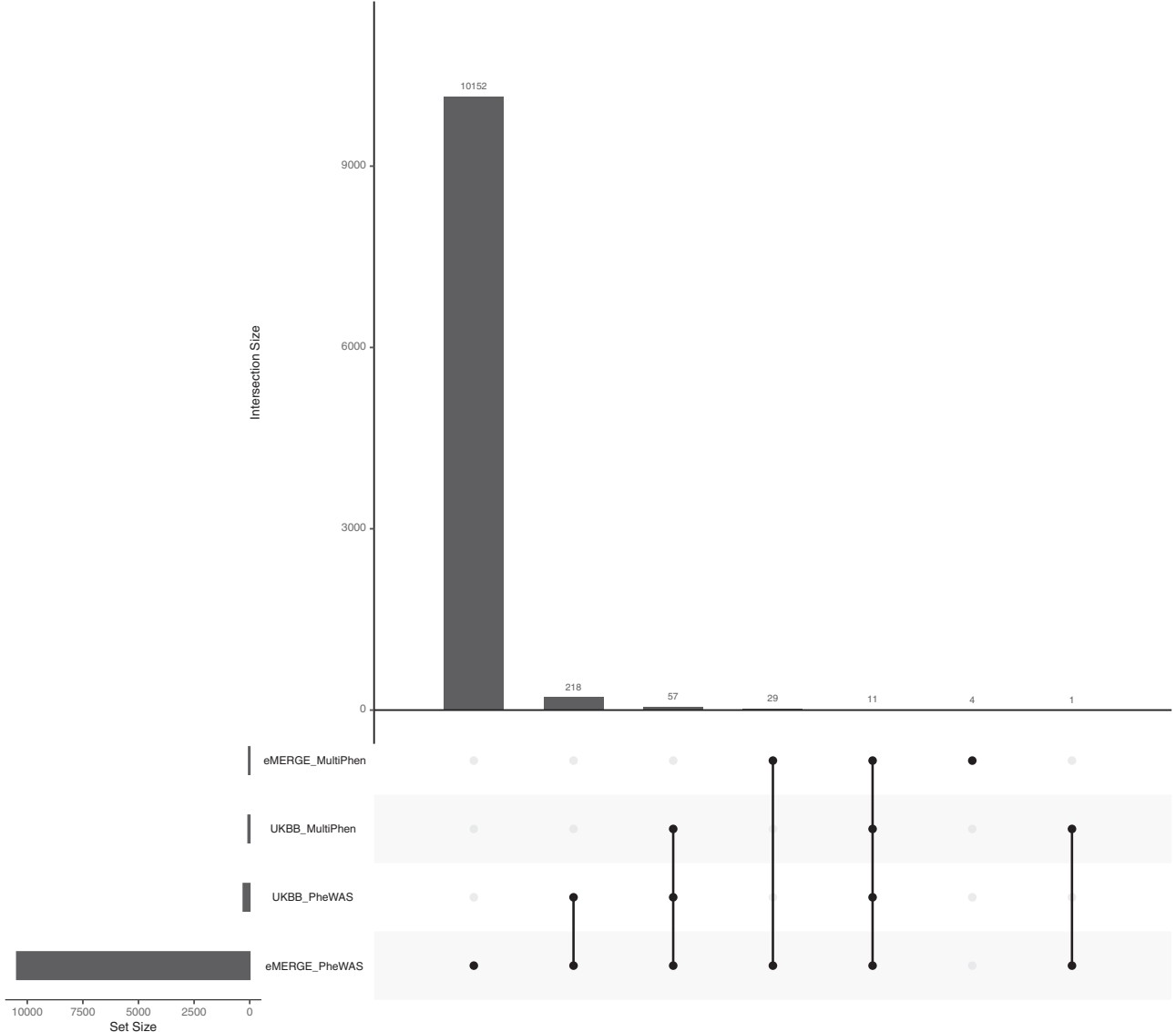

**Fig. 2 Comparison of the number of significant loci identified by PheWAS and MultiPhen from eMERGE and UK Biobank.** The $p$-value threshold is $1 \times 10^{-4}$. The number of loci are counted when they suggest significant associations with at least one phenotype. For PheWAS, we included the SNPs when its minimum $p$-value among phenotypes passed the threshold. Here, the total number of loci represents the independent loci we tested in both eMERGE and the UK Biobank.

Fig. 2B). Pleiotropic effects of these loci were formally tested as reported in the next section for both eMERGE and the UK Biobank.

**Formal test of pleiotropy**. The formal test of pleiotropy was conducted on the 11 loci (607 SNPs) using a $p$-value threshold of $1 \times 10^{-8}$ for a selected set of phenotypes in each of the two datasets, independently. There were 3 loci (52 SNPs) in eMERGE and 2 loci (59 SNPs) in UKBB that showed associations with *both* circulatory system diseases *and* nervous system disorders (Fig. 4; details in Supplementary Data 3). We characterized the direction of the genetic effects from the PheWAS results (Supplementary Data 4). An illustration of identified pleiotropic relationships among disease categories is shown in Fig. 5 (details in Supplementary Data 5). We reviewed the NHGRI-EBI GWAS catalog[37] for discovered pleiotropic common SNPs, and their associated traits relevant to our trait of interest and the direction of genetic effects are reported in Supplementary Data 3. We also discussed

the number of cases that overlap between traits as well as the correlation among traits in the Supplementary Note.

We identified rs157582 at chromosome 19q13.32 that suggested pleiotropy across circulatory system diseases and nervous system disorders from UKBB. There are 20 SNPs that suggested pleiotropy in the region (Supplementary Data 3, regional LD in Fig. 6). Those SNPs mapped to a region containing the genes *APOC1*, *APOC1P1*, *TOMM40*, *APOE*, and *NECTIN2*. All SNPs are associated with atherosclerotic heart disease, Alzheimer's disease, and dementia, while 14 SNPs are also associated with angina pectoris and 18 SNPs are associated with delirium. This region was found to be significantly associated with Alzheimer's disease in previous studies[38–40]. There are 8 SNPs that have previously demonstrated associations with cardiovascular disease risk factors such as HDL cholesterol, LDL cholesterol, total cholesterol, and triglycerides[41–44]. Only one SNP, rs4420638, has previously been associated with coronary artery disease[45] based on our review of the NHGRI-EBI GWAS catalog[37]. Our study showed the associations of these

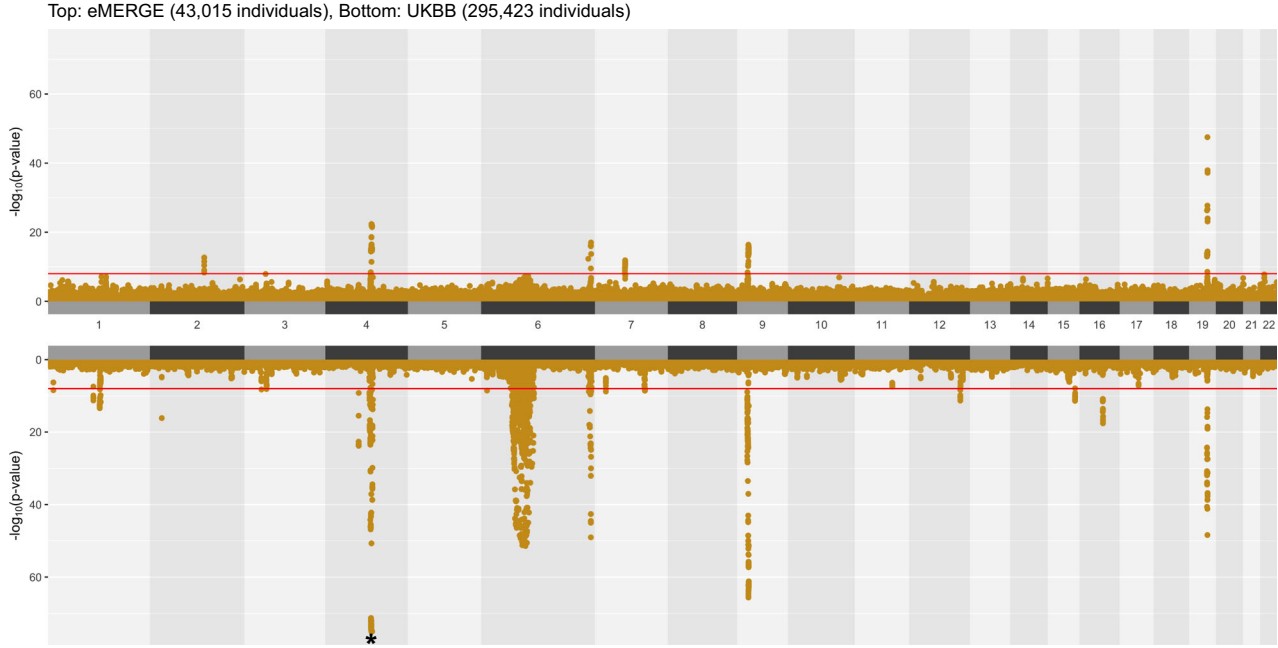

**Fig. 3 Landscape of MultiPhen results.** A position-to-position comparison of MultiPhen results between eMERGE and UKBB. The red line indicates a *p*-value of $1 \times 10^{-8}$. To reduce the margin induced by the extreme small *p*-values, we have collapsed SNPs with *p*-value < $1 \times 10^{-75}$ into one overlapping circle indicated by an asterisk on chromosome 4 for UKBB.

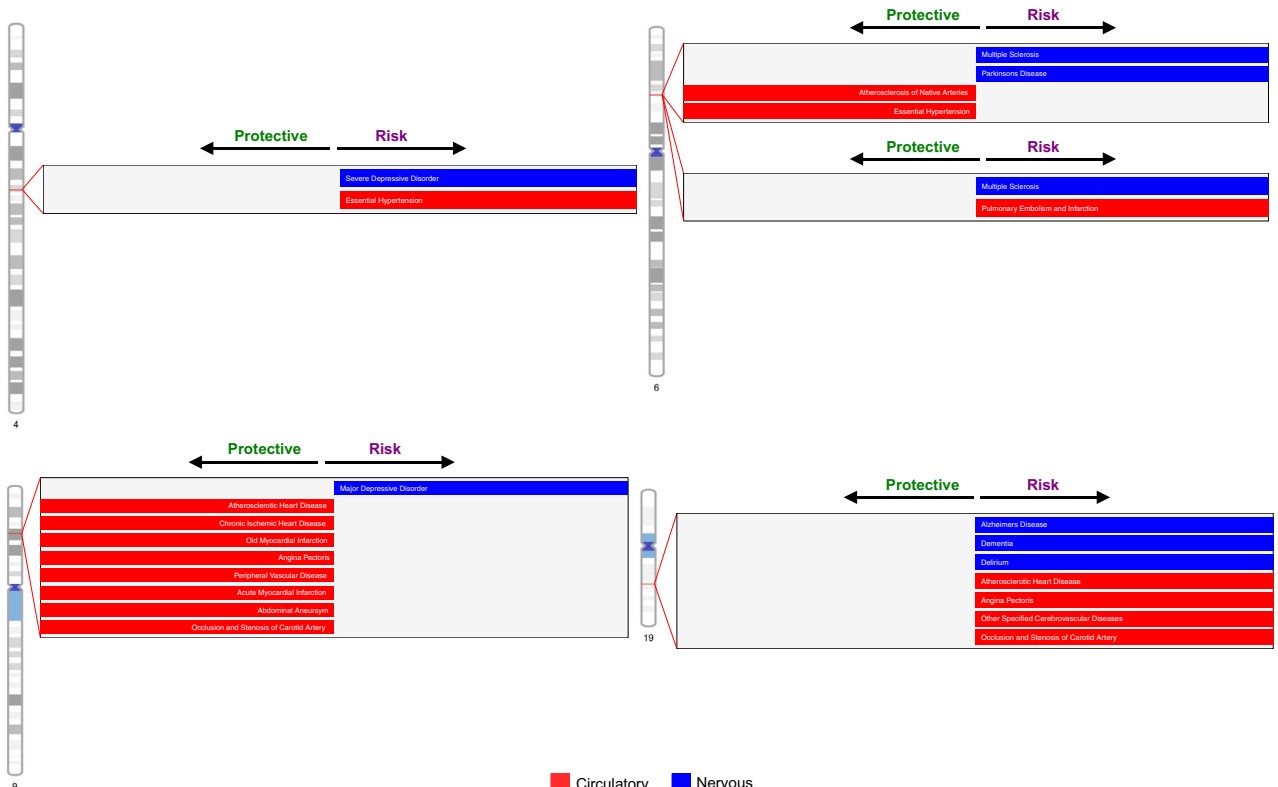

**Fig. 4 Characterization of top associated diseases for identified pleiotropy.** The diseases are characterized by sequential multivariate analyses and the direction of genetic effect is obtained from PheWAS results. The direction of genetic effect is based on the tested allele in our study. More details are shown in Supplementary Data 3. Note that the direction of genetic effect on chromosome 9 is a mixture of risk and protective effect for our tested alleles on two disease categories but overall opposite directions.

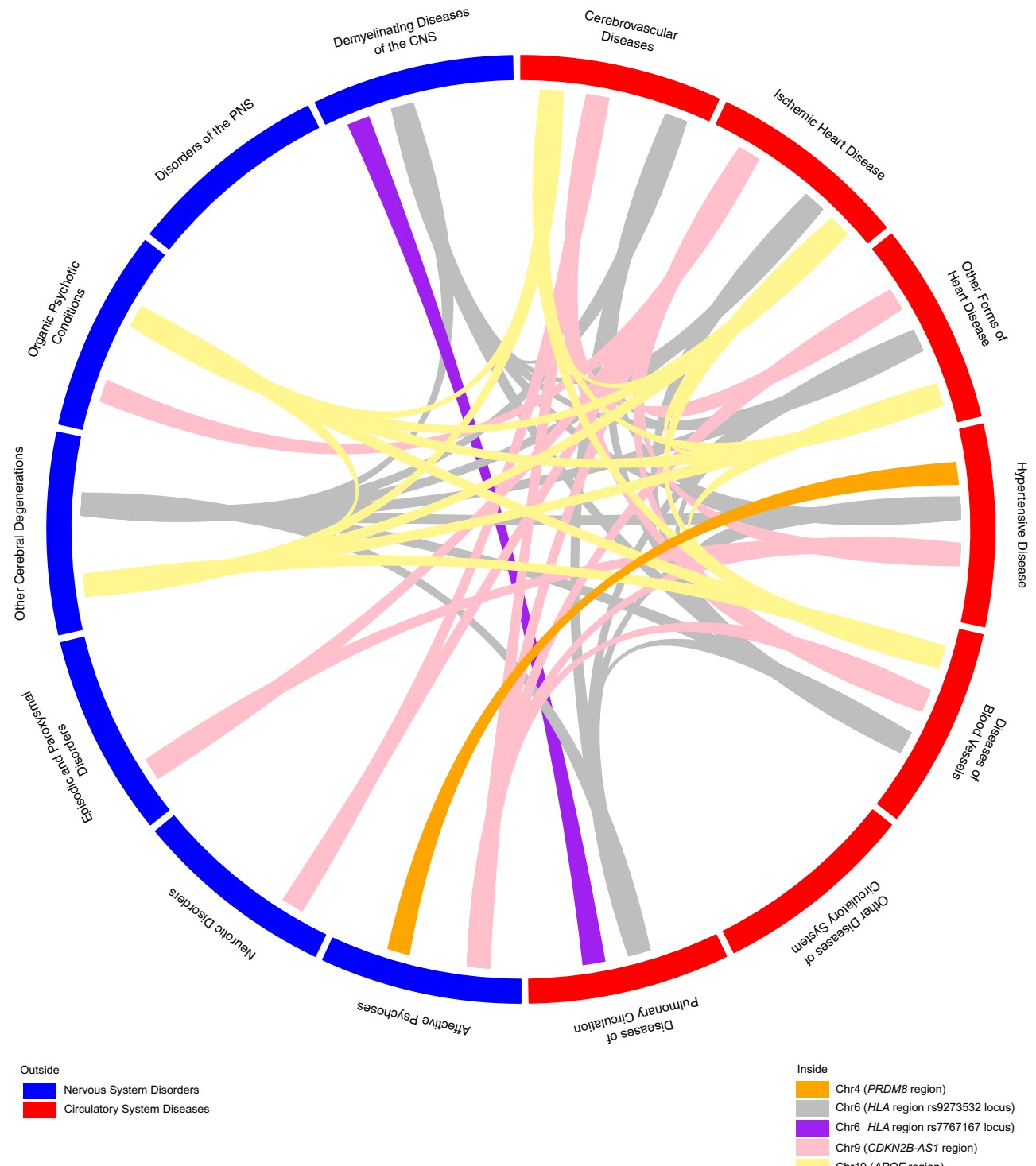

**Fig. 5 Disease relationships linked by pleiotropic SNPs.** The results are obtained from sequential multivariate analyses for both eMERGE network and UK Biobank. We demonstrate pleiotropy among disease categories by connecting them using SNPs that are significantly associated with at least one nervous system and one circulatory system disease category.

SNPs with additional circulatory system disease phenotypes such as acute transmural myocardial infarction of inferior wall and occlusion and stenosis of carotid artery. All of the SNPs demonstrated risk pleiotropic effects across all the identified circulatory system diseases and nervous system disorders, which is consistent with suggested trait-related associations from previously published studies in the GWAS catalog (Supplementary Data 3). Based on the evidence in the literature, the

chromosome 19 results are predominantly positive control associations that confirm previous findings (thus, these are proof-of-concept signals).

We identified locus rs10811656 (63 SNPs) at chromosome 9p21.3 that demonstrated pleiotropic associations with a wide range of circulatory system diseases and major depressive affective disorders from the eMERGE and UKBB (Supplementary Data 3, regional LD in Supplementary Fig. 3A). The SNPs

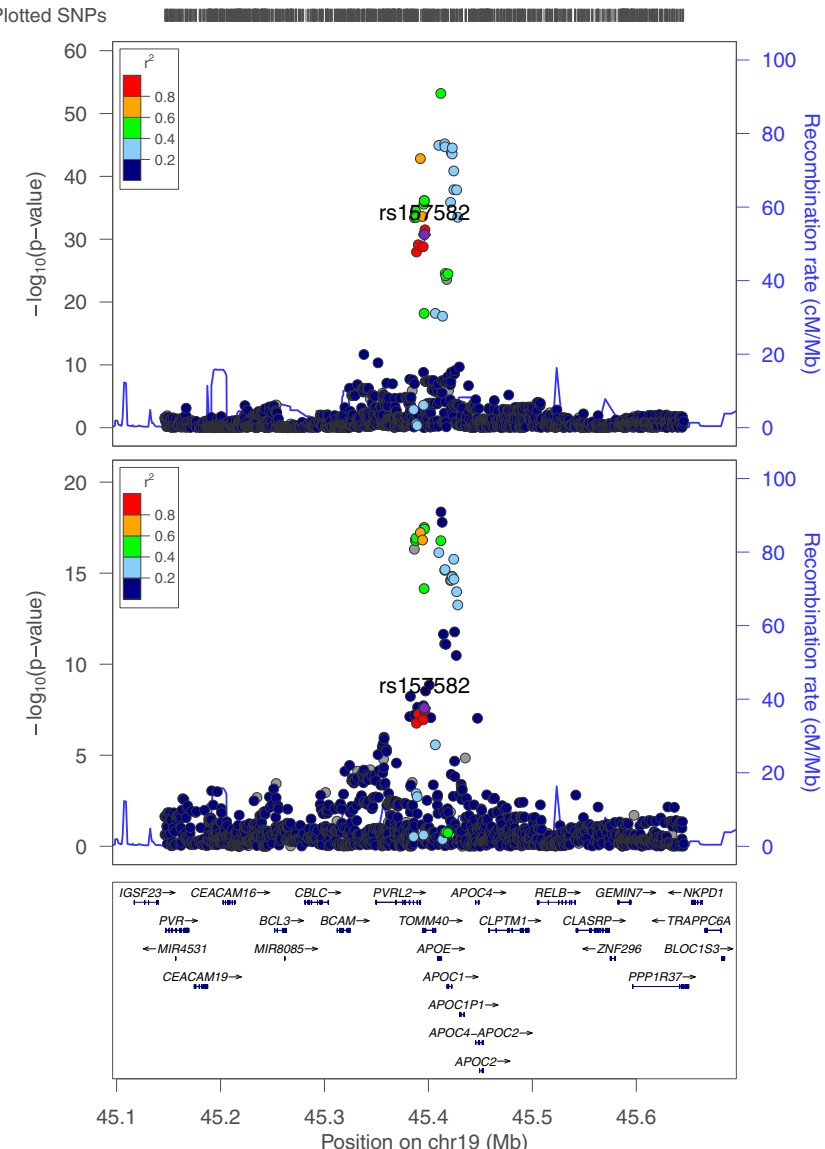

**Fig. 6 Regional LD relationships near rs157582 locus on chromosome 19 from UKBB.** The phenotype in the top plot is Alzheimer's disease; and the bottom plot is atherosclerotic heart disease.

mapped to the *CDKN2B* antisense RNA 1 region, which has long been known as a hot spot that is associated with cardiovascular diseases[46]. We not only detected previously known SNPs associated with cardiovascular diseases, such as rs10757278[47] and rs1333045[48], but also demonstrated a novel potential pleiotropic effect on major depressive disorders in this region, which was not observed in the GWAS catalog. Most of the SNPs were found to have opposite directions of genetic effect on circulatory system diseases and major depressive disorders (Supplementary Data 3); an example of antagonistic pleiotropy. For SNPs previously known to be associated with circulatory system diseases, the direction of genetic effects was consistent with previous studies in the GWAS catalog (Supplementary Data 3).

We characterized two loci rs9273532 and rs7767167 that have suggested pleiotropy on chromosome 6 near the *HLA* complex region at 6p21.3 in eMERGE (Supplementary Data 3, regional LD in Supplementary Fig. 3B, C). The genetic variants near locus rs9273532 showed novel pleiotropic associations with atherosclerosis of arteries of extremities, multiple sclerosis, and Parkinson's disease (Supplementary Data 3), none of which have

been reported in the GWAS catalog (though there are other SNPs in the HLA region that have previously been associated with multiple sclerosis[49,50]). SNPs near the rs9273532 locus demonstrated opposite directions of effect on circulatory system diseases and nervous system disorders (Supplementary Data 3). Similarly, there are 9 SNPs that were identified near rs9273532 locus (*LOC101929163/NOTCH4* region) in the UK Biobank, which are in high LD, have opposite directions of effect on essential hypertension and multiple sclerosis, which also has not been characterized before in the GWAS catalog. SNPs near the rs7767167 locus showed the same direction of effect (risk effect of tested allele) on pulmonary embolism and infarction and multiple sclerosis.

Finally, we also identified the rs16998073 locus (3 SNPs) near *PRDM8/FGF5* at chromosome 4q21.21 that are associated with essential hypertension and severe depressive episode with psychotic symptoms from UKBB, with a risk genetic effect on both diseases (Supplementary Data 3, regional LD in Supplementary Fig. 3D). All SNPs were suggested in the studies from the GWAS catalog to increase the risk of hypertension or related traits[51-56] (positive controls in our study), but we did not find any

evidence that they increase the risk of severe depressive disorders in the literature, thus potentially novel pleiotropy.

## Discussion

Many clinical and epidemiological studies have suggested the co-occurrence of circulatory system diseases and nervous system disorders. However, the genetic contributions to this relationship are largely unknown. To bridge this knowledge gap, we have characterized pleiotropy across these two broad disease categories by applying an effective analytical framework on two biobank cohorts: eMERGE and UKBB. Even though the prospective UKBB cohort has a large overall sample size, the case number for specific disease phenotypes is overall comparable to the medical eMERGE Network in most scenarios (Supplementary Data 1).

One of the advantages of our analytical design is the application of standardized univariate PheWAS and multi-trait joint analyses on two independent large datasets. As the availability of summary statistics from the GWAS catalog continues to increase, our ability to compare the summary statistics from univariate analyses, which is the commonly used approach to characterize pleiotropy, will continue to grow. However, multivariate methods, which have demonstrated generally greater power in simulation scenarios[57], have not been widely applied to natural biomedical datasets to study pleiotropy among disease states. The primary reasons are that most multivariate analyses in general are characterized by the following: 1. Most require individual-level data; 2. Are computationally intensive, and 3. Only test a null hypothesis that a variant affects none of the phenotypes examined (rather than identifying which subset of phenotypes are associated). We have addressed these challenges by obtaining individual-level data, splitting the genotype file into small chunks and running the analyses in parallel, and we have conducted a formal test of pleiotropy to pinpoint the specific associated phenotypes. We have applied both univariate PheWAS and multi-trait joint analyses as complementary methods to provide supporting evidence for our findings and identify a smaller set of SNPs to explore a formal statistical test of pleiotropy. Subsequently, there are multivariate methods, such as MTAG[16] or MultiABEL[58], that perform multi-trait analysis using GWAS summary statistics in a more computationally efficient manner. However, some of the summary-statistics based methods treat sample overlap as a nuisance and correct for it, while also being unable to consider scenarios where an individual has multiple phenotypes diagnosed. This is an additional motivation for using a method, like Multi-Phen, that handles the scenario when a phenotype has been collected for the same set of individuals – e.g. EHR linked biobank datasets. Summary-statistics based methods are generally computational efficient and preserving data privacy, while the accuracy or precision is often reported to be lower compared to individual-level data based methods[25,59]. Since we had access to individual-level data, we chose to use MultiPhen for our multi-trait analyses.

We characterized 11 loci (607 SNPs) that were identified by both PheWAS and MultiPhen methods in the discovery analyses eMERGE and replicated in UKBB (Supplementary Data 2). The SNPs in these 11 loci were associated with at least one tested phenotype. However, the definition of pleiotropy requires a genetic variant to influence more than one phenotype. Therefore, we have identified the precise set of phenotypes associated with a SNP via the sequential multivariate method (a formal test of pleiotropy). To assist the interpretation of pleiotropy, genetic effect sizes were collected from univariate PheWAS results. Additionally, the evaluation of the proportion of case overlap and conditional analyses on each identified phenotype set indicate that our discovered pleiotropy signals are likely genetic

associations of potential pleiotropy rather than due to comorbidity between circulatory system diseases and nervous system disorders (see Supplementary Note and Supplementary Fig. 6).

SNPs that were identified on chromosome 19 were previously known to increase the risk of Alzheimer's disease and cardiovascular disease risk factors from GWAS catalog[60] (proof of concept findings). We have identified consistent pleiotropic effects in this region with multiple cardiovascular disease states such as atherosclerotic heart disease, left ventricular failure, occlusion and stenosis of carotid artery, and acute transmural myocardial infarction. The associations with atherosclerotic heart disease, Alzheimer's disease and dementia were found in both combined analyses and sex-stratified analyses (see Results and Supplementary Data 5–7). The decreased cerebral blood flow due to atherosclerosis is known to be associated with pathogenesis of Alzheimer's disease[61]. Roher et al. found increased cerebral artery occlusion and stenosis as a consequence of severe atherosclerotic heart disease in Alzheimer's disease from 54 consecutive autopsy cases. Moreover, reducing cardiovascular disease risk offers opportunities for intervention for Alzheimer's disease[62]. Understanding the disease mechanisms of pleiotropic genes, like this region on chromosome 19, will inform disease treatment options.

We observed an association based on SNPs near CDKN2B-AS1, which is associated with cardiovascular diseases, with the opposite genetic effect on the phenotype of severe depressive episode without psychotic symptoms. Although we did not identify any significant associations between CDKN2B-AS1 and major depressive disorders in the GWAS catalog, a recent bivariate scan study suggested that the genetic variants near CDKN2B-AS1 have the opposite effect on type 2 diabetes and major depressive disorders[63]; this confirms our findings. A recent study on 2,743 individuals suggested that coronary artery disease and obesity occur in patients with depression treated by selective serotonin reuptake inhibitors (SSRIs, antidepressant)[64]. The potential antagonistic pleiotropic effect of CDKN2B-AS1 might explain the occurrence of coronary artery diseases in patients treated for depression.

We have identified novel pleiotropy signals based on genetic variants near the HLA locus that are associated with atherosclerosis of arteries of extremities, multiple sclerosis, and Parkinson's disease, with opposite genetic effects on the circulatory system and nervous system diseases. Our discovered SNP associations have not been reported before. The HLA gene region, though, has been previously associated with multiple sclerosis and Parkinson's disease[65,66]. Moreover, it has been recognized that inflammation is involved in atherosclerosis and coronary artery disease[67,68], thus highlighting the possible importance of autoimmune mechanisms and HLA polymorphisms. The SNPs near the HLA (NOTCH4; LOC101929163) region demonstrated association between essential hypertension and multiple sclerosis, with opposite direction of genetic effect. The association was also seen in the female-only analyses (see Supplementary Note). We have not identified associations of our identified SNPs with hypertension or related traits and multiple sclerosis from the GWAS catalog, although SNP rs9267992 has been suggested to be associated with multiple sclerosis by one early GWAS study on 978 cases and 883 group-matched controls[66].

The SNPs we report near PRDM8/FGF5 on chromosome 4 showed pleiotropic risk associations with essential hypertension and severe depressive episode with psychotic symptoms. While these variants have previously been associated with hypertension or related traits such as diastolic and systolic blood pressure (per the GWAS catalog), they have not, to our knowledge, been associated with depressive disorders. Previous epidemiological studies have consistently shown an increased risk of hypertension in patients with depression and vice versa[69–71]. Our observed

novel pleiotropic associations might contribute to the explanation of the relationships between these diseases.

We acknowledge that we only characterized pleiotropic common variants in individuals of European-ancestry due to statistical power considerations. Future research on rare genetic variants as well as both common and rare variants in other ancestries will shed more lights on the shared biology between these classes of diseases. We also acknowledge that our manuscript reported statistical pleiotropy, which might include spurious pleiotropy which occurs when one tag SNP captures multiple casual variants or genes in high LD[4], such as can be seen in the *HLA* region. It is challenging to distinguish it from biological pleiotropy and caution should be taken into consideration of the possible underlying mechanisms driving the potential pleiotropy. Another limitation of our analyses is that we only tested a set of phenotypes for the sequential multivariate model using a univariate $p$-value ≤ 0.01 in each dataset, which resulted in different phenotypes tested between datasets and thus the formal test of pleiotropy was not an exact replication. The reason behind the selection of phenotypes is the drastically increased computational time as the number of associated phenotypes increases. For example, SNP rs1333046 that is associated with 20 phenotypes detected by sequential multivariate model in UKBB costs 587 h of CPU time. It currently would not be feasible for us to conduct sequential multivariate analyses for over 100 phenotypes. Future development of more computationally efficient methods would greatly facilitate the detection of pleiotropy.

We have characterized pleiotropy across circulatory system diseases and nervous system disorders by applying a combination of univariate, multivariate, and sequential multivariate methods on eMERGE and UKBB datasets. Our results have provided new insights into the genetics underlying the relationships between these disease categories, which may assist in future disease prevention and treatment. Our integrative analytical framework can also be applied to other disease categories to study pleiotropy comprehensively.

## Methods

This study was conducted under all relevant ethical regulations. UK Biobank was approved under application ID 32133. eMERGE study was approved by the eMERGE Network Publications Committee.

**Biobank datasets**. The eMERGE Phase III dataset contains high-density genotype data for 99,185 subjects coupled with longitudinal electronic health records (EHRs). Subjects were genotyped across 78 genotype array batches and imputed to ~40 million variants[72]. Details of the imputation have been discussed elsewhere[72]. Among 12 contributing study sites across the United States, we have included six adult study sites in this study: Marshfield Clinic Research Foundation, Kaiser Permanente/University of Washington, Vanderbilt University Medical Center, Mayo Clinic, Geisinger, and Partners Healthcare. The eMERGE dataset was used for discovery analysis.

UKBB cohort release version 2 has deep genetic and phenotypic data on ~500,000 individuals across the United Kingdom. Individuals were genotyped on two similar types of genotype array across 106 batches and imputed to 96 million variants[73]. eMERGE network and UKBB have the same genome build, GRCh37/hg19. The replication analyses in UK Biobank was performed on the statistically significant SNPs from eMERGE ($p \leq 10^{-4}$ described more below) that were also present and passed QC in the UK Biobank dataset.

**Phenotype definitions**. The phenotypes were defined based on the International Classification of Diseases (ICD) diagnosis codes extracted from the EHR. Since the disease coding practices and regulations differ between the US and the UK, the composition and distribution of diagnosis codes are different. To maximize the phenotypic information, we have accordingly applied different, yet complementary strategies to the two datasets.

Since ICD-10 codes have added specificity compared to ICD-9 codes, we chose to convert ICD-10 codes to ICD-9 codes. For UKBB, we have only included individuals who had ICD-10 occurrences to retain its original collection of disease codes and because fewer data were available for ICD-9 codes in the UKBB and during a more remote time period than the ICD9-CM codes found in the eMERGE dataset. Because the disease diagnosis codes in UKBB were curated and represented

by the presence or absence of a certain ICD codes, this information was used to define case status; this means that if a person has a certain ICD-10 code present in the EHR, that person would be assigned as a "case" for that phenotype. If the person did not have that diagnosis code, he/she would be assigned as a "control". As for eMERGE, we have converted ICD-10-CM to ICD-9-CM codes using a combination of general equivalence mappings[74] and manual review. Because eMERGE offers longitudinal measures on diagnosis codes, we have applied a "rule of three" on ICD-9 codes to define case status. This means that if a person had three or more occurrences of a certain ICD-9 code in their EHR on different clinic visits, that person would be assigned as a "case". If a person had either one or two occurrences of a particular ICD-9 code, an "NA" status would be assigned. Finally, if a person did not have any occurrence of a particular ICD-9 code, a "control" status would be assigned for that phenotype. This approach was used to assign case status for all available phenotypes. One general caveat of EHR data in the eMERGE dataset is that the absence of certain disease diagnosis code for some individuals does not equal the absence of the disease, as the patients might get the medical care at another institution thus may not present in our datasets. This would bias results toward the null, thus we don't expect that this impacted our study in a significant way.

**Genotype quality control**. For the eMERGE dataset, we dropped imputed genotype array batches with a mean R-squared of imputation score < 0.3 as well as batches that had fewer than 50 samples[72]. We also excluded genetic variants with a mean R-squared of imputation score <0.3 calculated across batches. We used a combination of self-reported European ancestry and principal component analyses to extract individuals of European ancestry for inclusion. We applied genotype call rate and sample call rate of ≥ 99% and selected genetic variants with a minor allele frequency (MAF) ≥ 0.01. We excluded SNPs with Hardy-Weinberg Equilibrium exact test $p$-value below $1 \times 10^{-10}$. We dropped related individuals that were second-degree relatives or closer with pi-hat larger than 0.25. Since our phenotypes of interest are the late-onset nervous system and circulatory system diseases, we selected European ancestry adult individuals only with age ≥25 years old. After QC, there are 43,015 individuals and 7,629,801 SNPs included for analysis. We generated principal components (PCs) for the final set of individuals using high quality, common SNPs (with MAF ≥ 0.05 and R-squared ≥ 0.7)[72] and adjusted for the first two PCs in all subsequent association analyses based on the proportion of variance explained by the PCs. The projection of the first two PCs and the proportion of variance explained by the PCs are provided in Supplementary Fig. 4.

For quality control in the UKBB, we largely followed the protocols of a previous publication[73] and utilized information provided as part of the data release. We excluded poor quality individuals according to previous publication[73]. We dropped related individuals that were second-degree relatives or closer with pi-hat >0.25. We have also removed individuals who had sex mismatches between self-reported and genetically inferred sex. Genetic variants with an imputation info score <0.3 and MAF < 0.01 were excluded. European ancestry individuals were extracted using a combination of self-reported white British ancestry and principal component analyses[73]. Since age at recruitment for the UKBB cohort is 40–69[73], we did not apply any age filter. After quality control, there were 377,921 individuals and 9,505,767 SNPs available for analysis. After applying the above-described phenotype filtering, there were 295,423 individuals from UKBB that had ICD-10 codes documented in their EHR data. This was the final sample size for UKBB used in all subsequent analyses. We used the first 20 PCs that were provided by the data release for the association analyses[73].

**Association analyses**
*PheWAS*. We performed genome-wide PheWAS for 43,015 eMERGE individuals and 7,629,801 SNPs across a total of 147 circulatory system diseases and nervous system disorders via PLINK[75] v1.9 software. Logistic regression models were adjusted by age, sex, eMERGE study site, and the first two PCs. There were about 1 billion association tests conducted in this genome-wide PheWAS. Out of the 147 phenotypes evaluated, nine phenotypes did not converge using PLINK due to the small case number per study site. To address this, we performed the same logistic association tests for those nine phenotypes using PLATO[76]. The larger number of default iterations in PLATO successfully resolved the non-convergence issue. The loci are defined using LD pruning in PLINK with parameter "–indep-pairwise 100 5 0.1". From approximately 1 billion association tests, 11,822 loci (145,131 SNPs) were statistically significant with a $p$-value ≤ $1 \times 10^{-4}$ from either univariate and/or multivariate analyses in eMERGE; these SNPs were selected for replication in UKBB. From this set of SNPs, we performed PheWAS on 10,472 loci (134,363 SNPs) that passed quality control in the UKBB dataset (SNPs were either dropped during QC or not present in the UKBB dataset). To address the ambiguity of SNPs with MAF near 0.5 in each of the two datasets, we have flipped the direction of genetic effect sizes for 552 SNPs in UKBB that had (a) MAF ≥ 0.4 and (b) reference and alternative alleles switched in eMERGE network. In the UKBB PheWAS, the following covariates were included for adjustment: age, sex, genotyping array, and the first 20 PCs. For UKBB we also re-ran the associations with Townsend Deprivation Index (TDI) as an additional covariate; the results did not change and since we do not have TDI for eMERGE, we did not include it in the results reported.

*Multi-trait joint analysis.* For multi-trait joint analyses, we used the MultiPhen[23] R package to perform our analyses. MultiPhen tests the linear combination of phenotypes by treating SNPs as response variables, and phenotypes as predictor variables. It uses a proportional odds regression model to test for statistical association. As was done for the PheWAS described above, we performed a genome-wide MultiPhen analysis for eMERGE. The MultiPhen analyses in UKBB were performed the same set of 134,363 SNPs (see PheWAS Methods section). The same set of covariates described in PheWAS Methods section were used in the MultiPhen analyses. All of the phenotypes (including *both* circulatory and nervous system diseases) have been jointly analyzed in the MultiPhen model. Because the current version of MultiPhen is not able to deal with NA phenotypes, we imputed NA with 0.5 for the eMERGE phenotypes. The presence of an NA indicates that a person had at least one instance of the ICD code in their EHR. This leads to a greater likelihood that the person is a case rather than a control. In a previous pilot study, we performed a sensitivity analysis on significant SNPs to evaluate this imputation method in eMERGE; we found that it retained the same level of statistical significance as imputing to 0 or 1[77]. Thus, based on our previous study, we kept the imputation of 0.5 for NA. The time and memory for running MultiPhen increases with the sample size and the number of phenotypes. In order to run analyses efficiently, we parallelized our operations by dividing the genome into subset files (2000 variants per file for eMERGE and 500 variants per file for UKBB).

*Sequential multivariate analysis.* To evaluate which association show evidence of pleiotropy, the next step in our study was to perform a formal test of pleiotropy. We selected the sequential multivariate analysis using the 'pleio' R package[27] to perform this test for pleiotropy. 'Pleio' extended the multivariate analysis framework to sequentially test the hypothesis that $k + 1$ traits are associated with the genotype given the null that k traits are associated[27]. It characterizes the exact traits that are associated with the SNP while accounting for the correlation among the traits. Note that the alternative hypothesis for the general multivariate framework is that there is at least one phenotype being associated with the genotype, i.e., we would not know the exact associated traits. One of the limitations of the method is that it does not provide individual genetic effect estimates for each phenotype, thus, we utilized the genetic effect estimates from the PheWAS of each corresponding phenotype as our interpretation of the genetic discovery (such as in Fig. 4). We have conducted sequential multivariate analysis on a set of 607 SNPs. This set was derived from the list of SNPs that met a *p*-value threshold of $1 \times 10^{-4}$ in eMERGE PheWAS and/or MultiPhen AND replicated in UKBB at a *p*-value threshold of $1 \times 10^{-4}$ in the UKBB PheWAS and/or MultiPhen. The same set of covariates have been adjusted as described in the PheWAS Methods section. Since the number of sequential tests increases drastically as the number of associated phenotypes increases, we have performed our analyses on a subset of selected phenotypes. We selected this set of phenotypes based on the univariate PheWAS analysis results. Each phenotype that had a PheWAS *p*-value < 0.01 for each SNP was selected for the sequential multivariate test. The set of phenotypes tested can be different between the two datasets due to differences in univariate *p*-value for each SNP-phenotype pair. The *p*-value significance threshold for rejecting the null hypothesis in the sequential multivariate model was set at $1 \times 10^{-8}$, the same as the genome-wide significance level. This threshold was chosen due to the same number of association tests being potentially performed using a general multivariate framework and in a univariate GWAS study. In other words, the output phenotypes of 'pleio' would need to have a multivariate joint significance of $<1 \times 10^{-8}$ to reject the null hypothesis.

**Conditional analyses.** We performed conditional analyses on the whole set of phenotypes that are associated with each identified pleiotropic SNP (see Results section). We evaluated all pairwise combinations of the phenotypes, with one as the dependent variable while another one as independent variable. Specifically, we applied logistic regression on dependent variable while treating another phenotype as an independent variable, along with previously mentioned covariates. We evaluated the impact of adjusting for another phenotype on the significance of the SNP by measuring the log odds ratio of the *p*-value from two events: conditional analysis and independent analysis (without adjusting for another phenotype). The form of log odds ratio is $\log_{10}\left(\frac{\frac{p_c}{1-p_c}}{\frac{p}{1-p}}\right)$, where $p_c$ denotes the *p*-value from the conditional analysis and $p$ denotes the *p*-value from the independent analysis. We plotted the mean of log odds ratio (across SNPs in the same region) in heatmap, where the phenotype on each row denotes the dependent variable and each column denotes the phenotypes that were being adjusted in the conditional analysis (Supplementary Fig. 5). When the log odds ratio deviates from zero, it suggests that adjusting for that particular phenotype (independent variable) changes the significance of the association with the other phenotype (dependent variable), thus suggesting that the association (for certain SNP) between one phenotype is related to another phenotype. On the other hand, if the value is close to zero, it's likely that the SNP is independently associated with both phenotypes rather than affect one trait through influencing the other one.

**Case overlap calculations.** We obtained the number of overlapping cases between pairwise phenotypes of identified pleiotropy. Since the case sample size varies among phenotypes due to different disease prevalence, we plotted the proportion of overlapping cases, calculated as the number of overlapping cases divided by the total case sample size. We demonstrated this distribution in heatmap, where the phenotype in the row refers to the total case sample size used as the denominator when calculating the proportion (Supplementary Fig. 5).

**Sex-stratified analyses.** The rationale of sex-stratified analyses is the same as the combined analyses except that we stratified the analyses by gender in the eMERGE and UKBB. There are 22,129 female and 20,886 male individuals in the eMERGE; there are 161,296 female and 134,127 male individuals in the UKBB. We performed PheWAS followed by sequential multivariate analyses to characterize pleiotropy. The covariates that were adjusted were the same as before except that 'sex' was excluded. The *p*-value threshold was also the same: the tested phenotypes in sequential model were selected using a PheWAS *p*-value of 0.01, and the *p*-value threshold for sequential multivariate testing is $1 \times 10^{-8}$. We did not apply case number filtering in sex-stratified analyses.

**Data visualization.** The Hudson R package[78,79] was used for comparing association results from eMERGE and UK Biobank (Figs. 1 & 3). The Venn diagram (Fig. 2 and Supplementary Fig. 2B) was created by UpSetR[34]. The demonstration of pleiotropy among disease categories were presented in circos plots[80] (Fig. 5 and Supplementary Fig. 5). Regional LD plots were generated by locuszoom[81]. The heatmap were generated using heatmap.2 function in 'gplots' R package[82].

**Reporting summary.** Further information on research design is available in the Nature Research Reporting Summary linked to this article.

## Data availability

This project is under UK Biobank application ID 32133. The eMERGE data have been deposited in the dbGaP database under accession code phs001584.v1.p1 [https://www.ncbi.nlm.nih.gov/projects/gap/cgi-bin/study.cgi?study_id=phs000360.v3.p1]. The access of UK Biobank and eMERGE can be obtained by application. The raw biobank data are protected and are not available due to data privacy laws. The summary statistics generated from eMERGE and UK Biobank in this study are provided in the Supplementary Data 4.

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

## Acknowledgements

We would like to thank Daniel J. Rader, Yong Chen, Dana C. Crawford, and Li-San Wang for helpful discussion on this project. We would like to thank Rachal Kember and Scott M. Damrauer for providing the manually reviewed ICD-CM conversion map. This work was in part supported by P50GM115318-04S1. eMERGE Network (Phase III). This phase of the eMERGE Network was initiated and funded by the NHGRI through the following grants: U01HG8657 (Group Health Cooperative/University of Washington); U01HG8685 (Brigham and Women's Hospital); U01HG8672 (Vanderbilt University Medical Center); U01HG8666 (Cincinnati Children's Hospital Medical Center); U01HG6379 (Mayo Clinic); U01HG8679 (Geisinger Clinic); U01HG8680 (Columbia University Health Sciences); U01HG8684 (Children's Hospital of Philadelphia); U01HG8673 (Northwestern University); U01HG8701 (Vanderbilt University Medical Center serving as the Coordinating Center); U01HG8676 (Partners Healthcare/Broad Institute); and U01HG8664 (Baylor College of Medicine). UK Biobank. All data for this cohort pertained to project 32133 – "Integration of multi-organ imaging phenotypes, clinical phenotypes, and genomic data".

## Author contributions

The study was conceptualized and designed by X.Z. and M.D.R. Statistical analyses were conducted by X.Z., Y.V., and W.P.B. Data visualization was performed by X.Z. and A.M.L. Phenotype curation was conducted by T.G.D., A.V., and X.Z. Data acquisition for UKB was performed by Y.V., M.D.R., X.Z., and A.V. Data acquisition for eMERGE was performed by W.K.C., D.C, J.C.D, S.H, G.P.J, I.K, E.B.L, L.J.R, D.J.S, J.W.S, I.B.S., W.Q.W, and C.W. The manuscript was written by X.Z. and M.D.R. The interpretation of the results and the critical feedback on the manuscript were provided by all authors.

## Competing interests

M.D.R is on the scientific advisory board for Cipherome. D.J.R serves on Scientific Advisory Boards for Alnylam, Novartis, Pfizer, and Verve and is a founder of Staten Biotechnology. No competing interests are declared for any other co-authors.
