## [Peer Review File · Nature Communications]

Title: Large-scale genomic analyses reveal insights into pleiotropy across circulatory system diseases and nervous system disordersREVIEWER COMMENTS

Reviewer #1 (Remarks to the Author):

Zhang et al. performed a formal test of pleiotropy across 107 circulatory system and 40 nervous system traits in the eMERGE Network and UK Biobank data and reported five genomic regions with significant evidence of pleiotropy, where they observed region-specific patterns of direction of pleiotropy. Of the five regions, ApoE was previously reported. The search of pleiotropy by using individual-level data is of interest but there are several points, which the authors need to consider.

1. Above all, the aims of the study and its outcomes do not seem to match well. The authors intended to investigate pleiotropy as a potential mechanism, which could account for the frequently observed co-occurrence of two disease categories; however, they did not sufficiently examine whether or not the direction of genetic effects on individual disease categories can support their co-occurrence.

2. It is difficult for the reviewer and possibly readers to differentiate novel findings from replication findings in the manuscript. For instance, pleiotropy at the ApoE region had been previously reported and the present study has turned out to validate the findings, at least for the region, by using individual-level data.

3. The way of testing pleiotropy, which the authors claim “a formal statistical test of pleiotropy” is difficult to understand according to Fig. S1. It is preferable to provide a more detailed flowchart, in which the statistical thresholds taken at each step (method) should be clarified. The authors set an exploratory p-value at $\leq 1E-4$ for PheWAS in the discovery stage (eMERGE), but it is unclear how the authors have narrowed down the list and come to conclude that only five regions pass the formal statistical test, apart from Fig. 2. In Fig. 2, the authors appear to have tested various combinations of statistical methods instead of following a pre-determined flowchart.

4. In this context, the reviewer cannot see the numbers of SNPs described in a sentence (lines 151-153), “In eMERGE, 1,093 SNPs passed exploratory p-value threshold from both PheWAS and MultiPhen analyses, whereas there were 54 SNPs that only showed significance in MultiPhen analyses.” in Fig. 2. Please clarify this.

5. In the similar vein, the relations between PheWAS (ie, univariate analysis) and MultiPhen (multivariate analysis) and between these two methods and the sequential multivariate method need to be more clearly explained.

6. Some SNP-disease associations seem to be newly discovered, e.g., HLA locus on chromosome 6. Because no detailed information is provided for the SNP-disease associations except for Supplementary Tables 3 and 4, it is difficult to evaluate whether or not the genetic association is robust. Please present the relevant information, such as the effect size and p-value, for the most significant SNP in the individual regions, both for the eMERGE Network and UK Biobank data plus the combined datasets; this will allow us to see if the SNP-disease associations for two disease categories attain genome-wide significant level or at least are formally replicated in two independent datasets.

7. Robustness (or reproducibility) of pleiotropy should be more clearly demonstrated, since different phenotypes were tested due to the discrepancy of ICD codes between two datasets.

8. Considering the description that 90% of GWAS loci are pleiotropic (ref. 6,7) (line 32), the identified number of genomic regions with pleiotropy seems to be relatively small. Please examine whether this

number, i.e., a total of 5 regions, is higher than the case assuming random occurrence.

Reviewer #2 (Remarks to the Author):

Large-scale genomic analyses reveal insights into pleiotropy across circulatory system diseases and nervous system disorders

Xinyuan Zhang et al.

This study is to identify genetic variants that have pleiotropic effects on circulatory system diseases and nervous system disorders. The authors used two independent datasets (genotype and phenotype data from eMERGE Network and UK Biobank) to which PheWAS and MultiPhen analyses were applied, to characterise pleiotropy between 107 circulatory system and 40 nervous system traits. The authors reported five genomic regions that were shown to have significant pleiotropic effects. The authors argued that their findings can provide context for future prevention and treatment strategies.

I have a number of questions and comments that should be positively considered to improve the manuscript.

1. Line 144 and 477. It is not clear how did the authors fit those phenotypes. 1) Did they use two sets of multiple regressions, one with 107 circulatory system diseases and the other with 40 nervous system disorders for each SNP? 2) Did they use a multiple regression with the 147 traits altogether? Or, 3) did they use multiple sets of multiple regressions each with a pair of a circulatory system disease and a nervous system disorder? I think #3 would be a proper analysis for the study hypothesis, i.e. testing pleiotropy between circulatory system and nervous system traits.

2. Fig. S2. Given the Bonferroni correction for multivariate analyses, I would guess a single multiple regression with all the traits (regardless of circulatory system or nervous system traits) was used for each SNP in the multi-trait joint analyses. If so, was this really to test pleiotropic effects between circulatory system or nervous system traits? It seems it also included pleiotropic effects within circulatory system traits or those within nervous system traits, this might inflate the signals.

3. Line 176. What is the formal test of pleiotropy? It is quite confusing. It seems that the authors applied PheWAS and MultiPhen to select candidate SNPs that have pleiotropic effects between circulatory system and nervous system traits by applying an arbitrary threshold of p-value ($10E-04$). But, the MultiPhen analysis appeared to be problematic because the significance could be biased due to pleiotropic effects between traits within circulatory system or within nervous system. For selected SNPs based on these crude criteria, the authors tested if there were significant pleiotropic effects between a circulatory system and a nervous system trait (e.g. Atherosclerotic heart disease and Alzheimer's disease), which might be referred to as 'the formal test'. I am not sure if the formal test was done for every pair between circulatory system and nervous system traits.

4. The authors should check if SNP effects estimated in eMERGE Network data for circulatory system traits can significantly predict the phenotypes of a nervous system trait in UK Biobank (and vice versa) in the context of polygenic risk score prediction. The authors may want to use those SNPs in the five significant genomic regions for this risk prediction. This is important because 1) this can prove if the five genomic regions have genuine pleiotropic variants that have effects on both circulatory system and nervous system traits, and 2) risk prediction in independent dataset is relevant to the context of future prevention and treatment strategies.

5. Line 474. The authors used covariates to adjust confounding effects, which are age, sex, genotyping array, and the first 20 PCs. At least for the SNPs in the five significant genomic regions, additional covariates such as assessment centre, Townsend deprivation index (or socio-economic status) and BMI should be used to check if the main findings still hold.

6. Line 488. It is not clear why the phenotypic imputation with different values wouldn't have any effect. What is the mean and variance of the number of missing values for the phenotypes? The phenotypic imputation is crude. Why not using the mean?

7. Table S1. Please add # controls as well.

Reviewer #3 (Remarks to the Author):

Zhang et al. presented a series of univariate and multivariate GWAS results on circulatory and nervous system disorders, trying to claim insights into underlying pleiotropy. I have quite a few major comments stated below, meanwhile I am also confused regarding the novelty of this report. If the pleiotropic nature of five genomic regions is the main message, none of the region/loci seems to be novel, and the wide-spread pleiotropic effects are kind of known, e.g. as part of the results/conclusions by Watanabe et al. (2019) Nat Genet. I can see that there are specific things about particular types of diseases which we need further understanding, but to me this report does not provide enough further and clear insights.

Major points:

1. One of the biggest issues in the authors' presentation is the definition of locus or loci. Directly counting the number of significant SNPs is really not the proper strategy in presenting genome-wide association results. The large number of discovered/replicated/reported SNPs are mostly in strong LD, making it extremely difficult to assess the results. How many loci were discovered? How many independent genetic associations are there? See also several points below.

2. Another general issue in this study is the design. The story is presented in a contrast of eMERGE and UKBB results manner, but replication was claimed. It is not a proper discovery-replication protocol. What have been discovered? In the abstract it says 'five regions', are these the discoveries that were

passed onto replication? But the text says all SNPs with $p < 1e-4$ were considered in replication analysis? Confusing.

3. What are the dots and lines in Figure 2? If I understand correctly, this figure is presenting the number of overlapped SNPs across analyses. But again, most of the SNPs are in LD or namely representing the same loci, so the numbers in the figure does not directly reflect the real genetic architecture.

4. Line 151-155. It looks like besides the SNPs discovered by both univariate and multivariate analysis, only those unique to multivariate analysis are of interest, why? How many SNPs were discovered by only univariate analysis while missed by multivariate analysis?

5. When talking about power of univariate and multivariate analysis, it seems the same genome-wide significance threshold $1e-8$ was applied. Why isn't multiple testing more problematic for univariate analysis across many phenotypes? This should be properly addressed when comparing signals between univariate and multivariate tests.

6. Line 168-171. Again on the LD between SNPs, what does it mean by 'SNP-specific characteristics'? Is it supposed to be functionally related? The fact that LD pruning would remove SNPs does not justify the ignorance of LD in each locus. There are much better ways to dissect SNPs with different underlying effects in LD, e.g. GCTA-COJO (Yang et al. Nat Genet) and SOJO (Ning et al. AJHG). This is also related to Figure 6 - the plotted LD relationship says nothing about the underlying genetic effects. Some colocalization analysis could be considered as well.

7. Because of the problem of LD not being properly considered, I feel rather disappointed about the presentation of the so-called 'formal test of pleiotropy' in Figure 4 etc. Many SNPs were counted, but they are not independent. Figure 4 basically presents 4 loci, thus I don't see the proportions of SNPs in each phenotype (in very tiny font) make any practical sense.

8. From Figure 4 to 5, another issue started to arise - it is not surprising that multiple diseases are linked via some shared loci in the genome, so what are the general genetic correlations between these diseases? And how does the pleiotropic nature of each locus contribute to such genetic correlations? I think this is an essential point missed in this study. In line 274-275, 'unknown genetic contributions' was mentioned but no genetic correlation was investigated.

9. Line 204-205. Not limited to this piece, again, talking about numbers of SNPs within the same locus being associated with different phenotypes does not make sense - the numbers simply vary with arbitrary p-value thresholding and the SNPs are not independent. It could be a single underlying causal variant driving all 20 significant SNPs in LD.

10. Line 286-289. I don't think multi-trait analysis was only demonstrated in simulations. Plenty of literature were missed, which could also been seen in the two false points presented by the authors. MTAG (Turley et al. Nat Genet) and MultiABEL (Shen et al. Nat Commun) both require only GWAS

summary statistics, and they are certainly not computationally intensive as no individual-level data is required.

11. Line 294-295. Why applying both univariate and multivariate tests reduces false positives? Without careful investigation, I don't think this can be stated.

12. Line 375-360. For replication, the issue is not only about p-value threshold but rather on the question of what is being replicated? If a signal is replicated by a multivariate test, are the underlying genetic effects on multiple traits replicated with the same directions? The p-value does not tell us that.

13. Line 364-365. The time-consuming computation is because important summary-level methods in literature were missed.

REVIEWER COMMENTS

Reviewer #1 (Remarks to the Author):

Zhang et al. performed a formal test of pleiotropy across 107 circulatory system and 40 nervous system traits in the eMERGE Network and UK Biobank data and reported five genomic regions with significant evidence of pleiotropy, where they observed region-specific patterns of direction of pleiotropy. Of the five regions, ApoE was previously reported. The search of pleiotropy by using individual-level data is of interest but there are several points, which the authors need to consider.

1. Above all, the aims of the study and its outcomes do not seem to match well. The authors intended to investigate pleiotropy as a potential mechanism, which could account for the frequently observed co-occurrence of two disease categories; however, they did not sufficiently examine whether or not the direction of genetic effects on individual disease categories can support their co-occurrence.

The previous literature showing disease co-occurrence in early clinical and epidemiological studies **motivates** us to study pleiotropy across these two disease categories. The aim of our study is to investigate pleiotropy using large-scale biobank data to understand the genetic contribution to disease relationships and to determine if this might explain some of the disease co-occurrence patterns previously reported. While disease co-occurrence can result from increased genetic risk on both disease categories, the study of pleiotropy encompasses **all types** of genetic relationships with disease, including both antagonistic and synergistic pleiotropy. This means that our pleiotropy investigations could have resulted in identifying genetic regions with the same direction of effect for two disease categories, or opposite direction of effect. Our initial hypothesis was that we would identify primarily synergistic pleiotropy (same direction of effect). However, what we found was unexpected. In our findings, we describe regions on chromosome 4 and 19, which have an increased risk on both disease categories, thus, synergistic pleiotropy could be used to explain disease co-occurrence. However, for other regions, specifically chromosome 6 and 9, we identified signals with the opposite direction of effect. Despite the reality that disease co-occurrence was the motivation of our study, the analyses led us to some examples of synergistic pleiotropy and some examples of antagonistic pleiotropy, which we did not necessarily expect. To dig into these results a bit further, we investigated the disease case overlap and discussed it in the supplementary section *“Follow-up evaluation of the impact of phenotypic relationships”*.

2. It is difficult for the reviewer and possibly readers to differentiate novel findings from replication findings in the manuscript. For instance, pleiotropy at the ApoE region had been previously reported and the present study has turned out to validate the findings, at least for the region, by using individual-level data.

Thank you for this observation. We discussed the novelty of each region in the Results section and Discussion section in detail by reviewing previous literature and the NHGRI/EBI GWAS catalog. For the ApoE region, as we stated in the manuscript (Results section), it has been shown to be associated with Alzheimer’s disease and cardiovascular disease risk factors such as HDL, LDL according to the GWAS catalog. We also reported a SNP in ApoE that was known for coronary artery disease. We agree with the reviewer that this region was a validation and state the finding as a “positive control” in Lines 235-236. For the other novel regions, we put a

more detailed discussion of the novelty for each region in the main manuscript (Lines 264, 275, 382). One additional aspect of novelty in this study is that most of previous findings have been conducted in independent studies (e.g. GWAS studies), but our work provides evidence that using a unified analytical framework to formally characterize pleiotropy can be a fruitful endeavor.

3. The way of testing pleiotropy, which the authors claim “a formal statistical test of pleiotropy” is difficult to understand according to Fig. S1. It is preferable to provide a more detailed flowchart, in which the statistical thresholds taken at each step (method) should be clarified. The authors set an exploratory p-value at $\leq 1E-4$ for PheWAS in the discovery stage (eMERGE), but it is unclear how the authors have narrowed down the list and come to conclude that only five regions pass the formal statistical test, apart from Fig. 2. In Fig. 2, the authors appear to have tested various combinations of statistical methods instead of following a pre-determined flowchart.

Thanks for your comment. We've made the change to Fig. S1 to reflect what the reviewer has suggested. We did not test different combinations of methods. We performed the univariate PheWAS analysis and the MultiPhen analysis genome-wide in eMERGE. We then performed replication in UK Biobank by testing the variants that passed a p-value threshold of 1×10^{-4} in the eMERGE PheWAS and MultiPhen. What Figure 2 is showing is how many statistically significant SNPs overlap in the different analyses.

4. In this context, the reviewer cannot see the numbers of SNPs described in a sentence (lines 151-153), “In eMERGE, 1,093 SNPs passed exploratory p-value threshold from both PheWAS and MultiPhen analyses, whereas there were 54 SNPs that only showed significance in MultiPhen analyses.” in Fig. 2. Please clarify this.

Thank you for this point; we would be happy to clarify the interpretation of the UpSet plot. The 1,093 came from the number of overlapping SNPs between the MultiPhen and PheWAS analyses in the eMERGE network. From Fig.2, it would be three categories in which you see a black dot connecting the first and the last row. So that would be $607 + 436 + 50 = 1093$ SNPs. As for 54 SNPs that only showed significance in MultiPhen, it describes the SNPs that are unique to MultiPhen but not in PheWAS in the eMERGE network. So that would be $51 + 2 + 1 = 54$ SNPs. We added these formulas to the main text (Lines 169-170). In addition, we provided formulas for every value that we reported from Fig. 2 in the main text (Lines 149 and 172).

5. In the similar vein, the relations between PheWAS (ie, univariate analysis) and MultiPhen (multivariate analysis) and between these two methods and the sequential multivariate method need to be more clearly explained.

Thanks for the comments. We explained the specificity of each statistics model in the introduction to make the relations clearer between three methods (Lines 47, 55, 69).

6. Some SNP-disease associations seem to be newly discovered, e.g., HLA locus on chromosome 6. Because no detailed information is provided for the SNP-disease associations except for Supplementary Tables 3 and 4, it is difficult to evaluate whether or not the genetic association is robust. Please present the relevant information, such as the effect size and p-

value, for the most significant SNP in the individual regions, both for the eMERGE Network and UK Biobank data plus the combined datasets; this will allow us to see if the SNP-disease associations for two disease categories attain genome-wide significant level or at least are formally replicated in two independent datasets.

This is a very good point. To ensure that other studies can look at all of the summary statistics, we have added a new Supplementary Table 7 to the manuscript. The effect size and p-value for each SNP-trait pair has been provided in Supplementary Table 7 for both eMERGE and UK Biobank datasets. We also mentioned this in the manuscript (Line 200).

7. Robustness (or reproducibility) of pleiotropy should be more clearly demonstrated, since different phenotypes were tested due to the discrepancy of ICD codes between two datasets.

While the predominant ICD code versions were different between eMERGE and UK Biobank datasets, we used ICD codes for the phenotype definitions for both studies. To make the phenotypes comparable between the studies, we used ICD categories as provided by the ICD medication classification system. Thus, the phenotypes tested in our study actually are the same between the two datasets. We manually matched the ICD disease diagnosis codes between the two datasets including the knowledge of a clinical expert in our team (T. Drivas). We clarified this in the manuscript (Lines 94-96).

8. Considering the description that 90% of GWAS loci are pleiotropic (ref. 6,7) (line 32), the identified number of genomic regions with pleiotropy seems to be relatively small. Please examine whether this number, i.e., a total of 5 regions, is higher than the case assuming random occurrence.

The reference that the reviewer is referring to investigated all of the existing GWAS reports from all of the diseases being studied which includes 4155 GWASs for a total of 588 traits across multiple disease categories, whereas the number of SNPs we identified from our study came from only two disease categories (circulatory and neurological), and our discovered 5 regions were identified by two independent methods from two independent datasets, followed by a formal test of pleiotropy. These 5 regions were the association signals in which we had the most confidence given our study design and after corrections for multiple testing. In summary, although it is difficult to assess how different this number of regions is in comparison to random chance, we are fairly confident in our results. It is a very different comparison to the papers we referenced.

Reviewer #2 (Remarks to the Author):

Large-scale genomic analyses reveal insights into pleiotropy across circulatory system diseases and nervous system disorders
Xinyuan Zhang et al.

This study is to identify genetic variants that have pleiotropic effects on circulatory system diseases and nervous system disorders. The authors used two independent datasets (genotype and phenotype data from eMERGE Network and UK Biobank) to which PheWAS and MultiPhen analyses were applied, to characterise pleiotropy between 107 circulatory system and 40 nervous system traits. The authors reported five genomic regions that were shown to have significant pleiotropic effects. The authors argued that their findings can provide context for

future prevention and treatment strategies.

I have a number of questions and comments that should be positively considered to improve the manuscript.

1. Line 144 and 477. It is not clear how did the authors fit those phenotypes. 1) Did they use two sets of multiple regressions, one with 107 circulatory system diseases and the other with 40 nervous system disorders for each SNP? 2) Did they use a multiple regression with the 147 traits altogether? Or, 3) did they use multiple sets of multiple regressions each with a pair of a circulatory system disease and a nervous system disorder? I think #3 would be a proper analysis for the study hypothesis, i.e. testing pleiotropy between circulatory system and nervous system traits.

Thank you for the comments! We will clarify this here and have clarified it in the paper as well (Lines 518-520). We applied a multivariate regression with 147 traits altogether using MultiPhen, as the method tests the linear combination of the most associated phenotypes with the genotype. In theory, phenotypes that are not associated with the SNP would not contribute to the test statistics. We implemented this comprehensive test to ask the question: For each SNP, is the variant associated with one or more phenotypes? This allows us to compare MultiPhen to PheWAS. If you get a significant association in both PheWAS and MultiPhen for a given SNP, it really doesn't clarify whether multiple traits are involved as a single trait could be driving the significant MultiPhen association. If you get a significant MultiPhen association but a non-significant PheWAS association, it could mean that two or more closely related traits are acting in tandem to boost power of signal detection (or it could simply be an artifact of lower multiple comparison burden for MultiPhen). In this scenario, the null hypothesis of PheWAS and MultiPhen are consistent. Subsequently, we used PheWAS and MultiPhen to select the set of SNPs to then test for pleiotropy in a subsequent statistical analysis. We drew our conclusion of pleiotropy from the sequential multivariate method as described in the manuscript.

We agree with the reviewer that #3 would be a proper analysis if one wants to test all possible combinations of pairwise phenotypes. As a sensitivity check, we did compare the results from 'all phenotype MultiPhen' to 'pairwise MultiPhen' (one circulatory system and one nervous system phenotypes) in our pilot study and they remain significant at the same significance level in both analyses (PMID: 30864329). That said, we chose not to do a comprehensive pairwise analyses owing to the immense multiple comparison burden as well as because it still wouldn't help conclude with certainty if one or both traits are associated with the SNP. To determine pleiotropy, we would still need another method to provide such evidence, e.g. sequential multivariate method. Hence, we opted to conduct sequential analyses (formal test for pleiotropy) on a smaller set of pre-screened SNPs/traits instead. This is why we decided to use sequential multivariate as the final analysis in our pipeline.

2. Fig. S2. Given the Bonferroni correction for multivariate analyses, I would guess a single multiple regression with all the traits (regardless of circulatory system or nervous system traits) was used for each SNP in the multi-trait joint analyses. If so, was this really to test pleiotropic effects between circulatory system or nervous system traits? It seems it also included pleiotropic effects within circulatory system traits or those within nervous system traits, this might inflate the signals.

The answer to the reviewer's question is yes. All of the phenotypes were included in the MultiPhen analyses and the results from MultiPhen doesn't necessarily imply *only* pleiotropy. It

simply identifies all phenotypes that 'potentially' associate with the SNP. And yes, there could be within-circulatory or within-nervous system signals from MultiPhen results. That was the primary reason that we applied a formal test of pleiotropy following MultiPhen to truly identify which specific phenotypes were associated with the SNP and determine the most plausible examples of pleiotropy between the two disease categories our study is based upon. We only reported the regions where the SNPs are associated with at least one circulatory system and one nervous system phenotype, though we report both synergistic and antagonistic pleiotropy.

3. Line 176. What is the formal test of pleiotropy? It is quite confusing. It seems that the authors applied PheWAS and MultiPhen to select candidate SNPs that have pleiotropic effects between circulatory system and nervous system traits by applying an arbitrary threshold of p-value ($10E-04$). But, the MultiPhen analysis appeared to be problematic because the significance could be biased due to pleiotropic effects between traits within circulatory system or within nervous system. For selected SNPs based on these crude criteria, the authors tested if there were significant pleiotropic effects between a circulatory system and a nervous system trait (e.g. Atherosclerotic heart disease and Alzheimer's disease), which might be referred to as 'the formal test'. I am not sure if the formal test was done for every pair between circulatory system and nervous system traits.

The formal test of pleiotropy, also known as sequential multivariate analysis, performs multivariate analysis iteratively. We indeed first selected candidate SNPs that potentially have pleiotropic effect as the reviewer summarized above, and we then performed a formal test of pleiotropy. It is not done for every pair of phenotypes, instead, it performs multivariate analysis on a set of phenotypes iteratively. It tests the null hypothesis that $k+1$ traits are associated with the SNP, given that the null of k associated traits was rejected. We described the method in detail in the Methods section (Lines 534-537). So, instead of saying that the SNP is associated with a given pair of circulatory and nervous system traits, it gives us all the traits that are associated with the SNP. We then identified the SNPs that were associated with at least one circulatory and one nervous system trait and reported these in the manuscript as suggested evidence of pleiotropy. As one can see from our results (Figure 4), pleiotropic SNPs can be associated with more than one phenotype per disease category (within category associations), which provides a comprehensive view of the pleiotropic association via formal test of pleiotropy.

4. The authors should check if SNP effects estimated in eMERGE Network data for circulatory system traits can significantly predict the phenotypes of a nervous system trait in UK Biobank (and vice versa) in the context of polygenic risk score prediction. The authors may want to use those SNPs in the five significant genomic regions for this risk prediction. This important because 1) this can prove if the five genomic regions have genuine pleiotropic variants that have effects on both circulatory system and nervous system traits, and 2) risk prediction in independent dataset is relevant to the context of future prevention and treatment strategies.

We truly appreciate this comment and we thought it was a great idea. Even though our study design was not designed for PRS, we attempted to calculate a PRS using all of the SNPs in the five regions as the reviewer had suggested. In general, PRS works well on diseases that are polygenic – meaning that the cumulative effect from lots of SNPs (even genome-wide) that have small effects on their own, but can be combined to indicate disease risk. Many successful PRS studies have been conducted using a large number of SNPs from multiple large-scale datasets. We totally understand that the reviewer wants to evaluate the overall disease risk prediction across disease types, however, unfortunately, current PRS methods do not work well for only a

small set of SNPs per each trait. We used the univariate results from the eMERGE dataset (as a reference dataset) to calculate the PRS to evaluate in the UK Biobank (as a target dataset) for all of the trait combinations. We tried both LDpred and PRScise software; we ended up using PRScise as LDpred largely reduces the number of SNPs in the model because it cannot take SNPs with missingness. Because our selected SNPs were replicated among methods and datasets, we had 109 SNPs being tested for all traits combined, among which, an even smaller subset of SNPs were significantly associated with each trait after clumping and thresholding steps in PRScise. Unfortunately, the number of SNPs limited our power in calculating the PRS. The observed polygenic risk score is less than 5%, which makes sense given the low number of SNPs used in the PRS model – 30 models that have PRS 1-5% only uses 1 or 2 SNPs in the model. Our goal in this study is to discover the pleiotropic SNPs instead of evaluating the polygenic prediction power, which might do the best using genome-wide SNPs from large-scale biobank or from meta-analyses. In order to fully evaluate the prediction power among these traits, we would need to design a large-scale study that estimated the effect sizes and p-values from multiple datasets and use these PRS to predict disease risk on another dataset that has large proportion of overlapping SNPs between the reference and target datasets. This is especially challenging for the traits that being studied in this project, as our phenotypes were derived from the disease diagnosis code from the electronic health records – which may not be available for many datasets, and most of the published GWAS are focused on more prevalent diseases where the disease definitions are generally broader than using ICD codes. We can see this type of study turning into a very exciting project, with the focus on the genome-wide disease susceptibility or genetic background to predict risk for more prevalent cardiovascular and neurological diseases with large sample sizes across multiple datasets.

5. Line 474. The authors used covariates to adjust confounding effects, which are age, sex, genotyping array, and the first 20 PCs. At least for the SNPs in the five significant genomic regions, additional covariates such as assessment centre, Townsend deprivation index (or socio-economic status) and BMI should be used to check if the main findings still hold.

Thank you so much for your comment for including more covariates. The covariates that we selected for adjustment are based on what has become common practice in the GWAS literature; specifically, a UK Biobank manuscript has used the same set of covariates in their GWAS analyses (Bycroft *et al*). The same strategy has been used for a recent eMERGE GWAS analysis (Stanaway *et al*). We agree, however, with the reviewer that it is a good idea to perform sensitivity analyses to determine if our main findings hold with the addition of other covariates. For assessment centre, because the formal test of pleiotropy (pleio R package) cannot include ordinal covariates, we could not test for centre in the model due to the limitation of the software. As for BMI, there was a paper published in 2015 (Aschard *et al*, AJHG, cited 147 times) titled “Adjusting for Heritable Covariates Can Bias Effect Estimates in GWAS”, which they suggested that “unless we know with certainty that the tested variant does not influence the covariate, we recommend that the inclusion of such heritable covariates in the model should be avoided. Given evidence for a large number of pleiotropic genes across complex traits, it seems unlikely that any heritable covariates with a complex genetic architecture, e.g. BMI or WHR, will fulfill that condition.” We decided based on the literature, we should not include BMI as a covariate to avoid bias. That said, for Townsend deprivation index, we added it as an additional covariate in the UK Biobank and all the five regions still remain associated with both cardiovascular and neurological diseases. We have added this to the manuscript (Lines 507-509).

6. Line 488. It is not clear why the phenotypic imputation with different values wouldn't have any effect. What is the mean and variance of the number of missing values for the phenotypes? The phenotypic imputation is crude. Why not using the mean?

Thank you for raising this important point. The different imputation values having no effect is likely due to the definition of NA in the eMERGE network. The assignment of NA for a phenotype is when the individual has an ICD code in their EHR 1 or 2 times (see Phenotype Definitions section in the Methods). As described in the methods, we used a rule of three, which means that the ICD code must be present 3 times in order to be considered a case and zero times to be considered a control. So basically "NA" denotes a status when a person is likely to be a case 1 or 2 times, as compared to a case definition of at least 3 times. Whereas a control has no record of the code in their medical record. Since these individuals had the code at least one time in their medical record, we chose to use 0.5 to denote NA status for eMERGE. We have performed sensitivity analyses in our pilot study for eMERGE (*Zhang, PSB, 2019*, results from our previous sensitivity analysis are provided below). You can see that the significance remains at a similar level using different imputation methods. The mean proportion of missing values in the eMERGE is 3.27% and the variance is 0.11%. The low percentage of NA in our dataset might also contribute to the consistent significance level. There are many ways for phenotypic imputation in the literatures and it's beyond the scope of this manuscript to test all of them.

As far as why we did not use the mean, the mean would not be meaningful in these individuals. It is likely the situation that a person has one date with an ICD code in their chart and then many years of EHR data without the code. So, it is unclear how one would calculate a mean based on this binary code. The presence of the code is a 1, and the absence of a code is 0. This is very different from a quantitative trait where one could simply take the mean of whichever measurements are available in the EHR for that individual.

MultiPhen Tests	Impute NA as 0.5	Impute NA as 1	Impute NA as 0
1_36822024	6.84E-12	2.02E-11	4.41E-10
6_32569056	1.38E-11	9.36E-09	1.28E-11
14_106995720	5.12E-19	3.31E-19	5.78E-17
22_22876236	3.76E-11	5.86E-09	8.54E-09
22_22947156	9.56E-29	1.18E-27	3.69E-21
22_25420792	3.12E-41	1.31E-37	2.73E-28
22_25436904	5.77E-59	2.74E-56	5.97E-37
22_28250172	8.64E-23	6.21E-22	9.71E-15
22_33079917	2.51E-24	2.55E-23	5.72E-16

7. Table S1. Please add # controls as well.

Sure, thank you for the comment. The number of controls has been added to the Table S1.

Reviewer #3 (Remarks to the Author):

Zhang et al. presented a series of univariate and multivariate GWAS results on circulatory and nervous system disorders, trying to claim insights into underlying pleiotropy. I have quite a few

major comments stated below, meanwhile I am also confused regarding the novelty of this report. If the pleiotropic nature of five genomic regions is the main message, none of the region/loci seems to be novel, and the wide-spread pleiotropic effects are kind of known, e.g. as part of the results/conclusions by Watanabe et al. (2019) Nat Genet. I can see that there are specific things about particular types of diseases which we need further understanding, but to me this report does not provide enough further and clear insights.

Major points:

1. One of the biggest issue in the authors' presentation is the definition of locus or loci. Directly counting the number of significant SNPs is really not the proper strategy in presenting genome-wide association results. The large number of discovered/replicated/reported SNPs are mostly in strong LD, making it extremely difficult to assess the results. How many loci were discovered? How many independent genetic associations are there? See also several points below.

We appreciate this comment and will do our best to clarify. In the manuscript, when we mention "loci" in the introduction – the term was directly used in the manuscript that we cited (Line 32). Throughout the rest of the manuscript, we reported the number of SNPs and their LD structure because we don't want to miss any SNPs that have a unique association but may not have been analyzed after being LD pruned as part of the data preparation steps. Meanwhile, we did provide SNPs after LD pruning in each region in the Supplementary Table 2 for the reference. We found that it was important to report the SNPs identified directly in our manuscript, as it provides a more detailed picture about the association for other studies to look for additional replication. That said, we did report that our results are really based on only 5 genomic regions where the significant SNPs mapped to.

2. Another general issue in this study is the design. The story is presented in a contrast of eMERGE and UKBB results manner, but replication was claimed. It is not a proper discovery-replication protocol. What have been discovered? In the abstract it says 'five regions', are these the discoveries that were passed onto replication? But the text says all SNPs with $p < 1e-4$ were considered in replication analysis? Confusing.

We apologize for the confusion and we hope that the design and our language describing it is clearer now. It is true that we did not follow the traditional discovery analysis passing only significant results to replication. Instead, we followed a traditional discovery-replication protocol in Analysis 1, where we performed the PheWAS and MultiPhen genome-wide in eMERGE and then only tested the SNPs with a p-value $< 1 \times 10^{-4}$ in UK Biobank. This stage is traditional discovery-replication. Based on this analysis, 607 SNPs were associated in eMERGE at this modest significance level and subsequently replicated in UK Biobank at the same nominal p-value threshold of 1×10^{-4} . Subsequently, we performed the sequential multivariate analysis on the 607 SNPs using both eMERGE and UK Biobank. Here, we looked for which signals showed evidence of pleiotropy at a p-value threshold of 1×10^{-8} in either dataset. The five regions that were identified using a formal test of pleiotropy on the 607 SNPs demonstrate the regions that had significant association with at least one circulatory system disease and one nervous system disorder. We hope the above explanations help with the understanding of the study design. We added p-value threshold in Line 195 and also in Figure S1 for the formal test of pleiotropy to make it clearer.

3. What are the dots and lines in Figure 2? If I understand correctly, this figure is presenting the

number of overlapped SNPs across analyses. But again, most of the SNPs are in LD or namely representing the same loci, so the numbers in the figure does not directly reflect the real genetic architecture.

Your understanding is correct. This UpSet plot represents the number of overlapping SNPs across the different analyses. While you are correct that it does not account for the LD in the regions that define these “loci”, we still think it is informative to look at the overlap of the SNPs in the results. All downstream analyses are focused on the loci, or genomic regions, where these association signals are focused so that we can begin to consider genetic architecture more specifically. Still, since we are providing all summary statistics from the analyses, it is useful to look at the overlap of association signals across the different analyses.

4. Line 151-155. It looks like besides the SNPs discovered by both univariate and multivariate analysis, only those unique to multivariate analysis are of interest, why? How many SNPs were discovered by only univariate analysis while missed by multivariate analysis?

This is a very important question and it clearly points out to us that these UpSet plots are more challenging to interpret than we had appreciated. Figure 2, which is an UpSet plot, points out the unique SNPs to univariate analysis, unique SNPs to multivariate analysis, and the overlap of both analyses. In the text of the manuscript, we were pointing out the results from the multivariate model because the univariate method (PheWAS) is a popular, commonly used method in many other publications; as such, we wanted to point how the multivariate approach produces different results from the univariate method. This is another reason to show all of the SNPs, rather than just the loci. The detailed comparison is shown in Figure 2. Readers can easily get whichever comparisons that they are interested in from Figure 2. Because multiple reviewers were confused by the UpSet plot, we have added additional formulas to clarify how to interpret the reported values (Lines 149, 169-172).

5. When talking about power of univariate and multivariate analysis, it seems the same genome-wide significance threshold $1e-8$ was applied. Why isn't multiple testing more problematic for univariate analysis across many phenotypes? This should be properly addressed when comparing signals between univariate and multivariate tests.

The use of 1×10^{-8} was just the GWAS significant line in the Hudson plot for plotting purpose. We did realize the multiple testing issue and provided the comparison using Bonferroni threshold (customized by the number of tests for each method) in Supplementary Fig S2.

6. Line 168-171. Again on the LD between SNPs, what does it mean by 'SNP-specific characteristics'? Is it supposed to be functionally related? The fact that LD pruning would remove SNPs does not justify the ignorance of LD in each locus. There are much better ways to dissect SNPs with different underlying effects in LD, e.g. GCTA-COJO (Yang et al. Nat Genet) and SOJO (Ning et al. AJHG). This is also related to Figure 6 - the plotted LD relationship says nothing about the underlying genetic effects. Some colocalization analysis could be considered as well.

This is an interesting and complicated point. When one performs LD pruning across two datasets, which SNP is kept in the dataset from a set in LD is arbitrary. Thus, across two datasets (such as eMERGE and UK Biobank), different SNPs could be pruned out or kept in the dataset. Downstream analyses are then complicated as one looks for replication because there

are SNPs missing from each of the datasets due to pruning. To avoid this, we chose to not perform LD pruning in this step such that we would have the complete set of SNPs that are associated with the phenotypes in each region. They do not indicate multiple independent signals, merely, the set of SNPs in the LD region that are associated. When we refer to the “SNP-specific characteristics”, we simply mean the unique pattern of the SNP-phenotype associations in each region.

We chose to leave the number of independent associations at a locus open ended because of the drawbacks of existing approaches. For instance, GCTA-COJO can overcorrect and miss useful predictors, whereas SOJO uses LASSO, which conducts variable selection and might not be suitable in instances where predictor variables are highly correlated or if the trait is highly polygenic (methods such as elastic net might be more suitable here). So, given the drawbacks of existing approaches, we left it open-ended at the level of 5 novel loci. Figure 6 provided the LD structure of the region and we added the magnitude and direction of the genetic effects in Supplementary Table 7.

7. Because of the problem of LD not being properly considered, I feel rather disappointed about the presentation of the so-called 'formal test of pleiotropy' in Figure 4 etc. Many SNPs were counted, but they are not independent. Figure 4 basically presents 4 loci, thus I don't see the proportions of SNPs in each phenotype (in very tiny font) make any practical sense.

You are correct that the SNPs are not fully independent. To provide the capability for others to reproduce our results or look for replication of signals in further independent datasets, we included all SNPs in our analyses and in our results files (Supplementary Table 7). However, our goal is not specifically focused on the independent SNPs, but rather the independent genomic regions where we see evidence of pleiotropy. We felt that it was useful to be aware of the proportion of SNPs in a region to represent the relationship of the region and diseases, however, it is true that they are not independent and in fact, are partially due to which SNPs passed imputation and quality control, so may not be as informative as we had hoped. Nonetheless, our goal in Figure 4 is to see where we observe a consistent direction of genetic effect on two disease categories (synergistic pleiotropy) and where we see opposite direction of effect (antagonistic pleiotropy).

8. From Figure 4 to 5, another issue started to arise - it is not surprising that multiple diseases are linked via some shared loci in the genome, so what are the general genetic correlations between these diseases? And how does the pleiotropic nature of each locus contribute to such genetic correlations? I think this is an essential point missed in this study. In line 274-275, 'unknown genetic contributions' was mentioned but no genetic correlation was investigated.

We appreciate your comments. We tried the most widely used software -- LD score regression (ldsc software) to evaluate the genetic correlation in the eMERGE dataset. As we performed the genome-wide analysis for the eMERGE dataset, we have genome-wide coverage to then compare to the UK Biobank. However, the ldsc package suggested that the sample size for eMERGE is too small such that it cannot give us a reliable estimation on the genetic correlation. There are other methods as well that can estimate genetic correlation for individual level data, but we suspect implementing them will result in the same problem, i.e. sample size is too small relative to the number of predictors. Most of studies in estimating genetic correlation uses estimates from a large meta-analysis such that it has a very large sample size. However, we do not have such meta-analyses available for all of the traits being explored in this manuscript.

Unfortunately, we believe it is beyond the scope of the current research to obtain these meta-analysis statistics for all of these traits; however, we can see it turn into another separate project by collecting all of the summary statistics for all of these phenotypes that were published by other studies. Still it would be important to note that the phenotypes will not be precisely the same as the phenotypes included in our study since our phenotypes were derived from the ICD codes from the electronic health records.

9. Line 204-205. Not limited to this piece, again, talking about numbers of SNPs within the same locus being associated with different phenotypes does not make sense - the numbers simply vary with arbitrary p-value thresholding and the SNPs are not independent. It could be a single underlying causal variant driving all 20 significant SNPs in LD.

We are simply reporting the total number of SNPs instead of LD-pruned SNPs at a locus. We do not claim that the reported SNPs are independent. We provided the LD information in Supplementary Table 2 and we explained our logic of not using LD in the main text (Lines 186-192).

10. Line 286-289. I don't think multi-trait analysis was only demonstrated in simulations. Plenty of literature were missed, which could also been seen in the two false points presented by the authors. MTAG (Turley et al. Nat Genet) and MultiABEL (Shen et al. Nat Commun) both require only GWAS summary statistics, and they are certainly not computationally intensive as no individual-level data is required.

The reviewer is correct that MTAG and MultiABEL are two methods using univariate GWAS summary statistics data, which are not computationally intensive; but these methods have less power compared to methods designed for individual level data according to the simulation studies. The biggest advantage of applying MultiPhen here is that we have access to measurements on multiple traits for the 'same' set of individuals and our estimates are hence cleaner. Meanwhile, summary-statistics based multi-trait analysis methods treat this 'sample overlap' as a nuisance and have to correct for it, which they do with varying levels of success. Since we have access to the individual level data for these datasets, it made more sense to use the well powered method for individual level data, specifically MultiPhen.

11. Line 294-295. Why applying both univariate and multivariate tests reduces false positives? Without careful investigation, I don't think this can be stated.

Thanks for your comment. We changed the wording to "provide supporting evidence for our findings".

12. Line 375-360. For replication, the issue is not only about p-value threshold but rather on the question of what is being replicated? If a signal is replicated by a multivariate test, are the underlying genetic effects on multiple traits replicated with the same directions? The p-value does not tell us that.

The replication simply indicated from the p-value, which characterized the discovery of disease associated SNPs. It is not appropriate to interpret genetic effect (or beta) from the multivariate model, which is also a weakness of multivariate methods. The p-value from multivariate model

can be used to reject the null hypothesis (explained in Lines 313-315), which provides a fair comparison to the univariate method – ensure a consistent alternative hypothesis for these two methods. In addition, we used univariate and multivariate methods in our first step to select potential SNPs, and the pleiotropy was characterized carefully using the formal test of pleiotropy in the second step. We also compared the genetic effect of our results with the GWAS catalog and the directions are consistent (Supplementary Table 3).

13. Line 364-365. The time-consuming computation is because important summary-level methods in literature were missed.

The context of the paragraph refers to the sequential multivariate model – a formal test of pleiotropy method (sequential multivariate model), for which we don't see any literature proposing summary-level analyses. I assume that the reviewer is talking about summary-level methods for multiple traits, and we mentioned several methods in our manuscript (Lines 49-50) and discussed their comparison to individual level method extensively (Lines 309-310, 320-325).

REVIEWER COMMENTS

Reviewer #1 (Remarks to the Author):

Zhang et al. revised the manuscript in response to the reviewer's comments and it appears to have been improved substantially. The reviewer has a few comments on point #1.

Since there are two types of effect direction, same and opposite, at the loci showing significant association with distinct (but similar combination of) phenotypes (or disease traits), we had better be careful about the interpretation. Even though the authors performed a statistical test of pleiotropy, is it still possible that not a single gene variant but multiple alleles at the gene of interest may be present, with the individual alleles exerting the opposite direction of effect on multiple phenotypes?

Alternatively, is it possible that multiple variants (SNPs) of adjacent but distinct genes may be located on the same haplotype in the associated region, ie, in strong LD, which could explain some of the finding of antagonistic pleiotropy? If so, such an observation should be not a pleiotropy but a composite of separate SNP-disease associations. Please discuss these possibilities.

Reviewer #2 (Remarks to the Author):

In line with Q1 in the previous report, I have a further question for the line 198 – 199, and for the authors' response "... we chose not to do a comprehensive pairwise analyses owing to the immense multiple comparison burden as well as because it still wouldn't help conclude with certainty if one or both traits are associated with the SNP".

It is still not clear if the significance of the 52 SNPs in eMERGE and 59 SNPs in UKBB are entirely due to pleiotropy between circulatory and nervous system disorders or due to something else (e.g. pleiotropy within circulatory (or nervous) system disorders) (see Figure 5). I am not quite sure that the threshold used in the formal test of pleiotropy can be justified well. I would suggest the authors should do comprehensive pairwise analyses with a proper multiple test correction and should discuss the difference in their results if there is significant difference.

For Q5, the authors can transform the ordinal or class variable to $n \times m$ matrix with 0 and 1 where n is the same size and m is the number of levels of the variable. Or, the phenotypes can be pre-adjusted for the variable before the main analysis. I don't see why this is not possible to test.

Finally, I wonder if the relationship between circulatory and nervous system disorders is mediated via genes involved in obesity. It would be useful to see how the results will be changed when the phenotypes should be adjusted for BMI. I am not sure if the issue raised in the paper (Adjusting for Heritable Covariates Can Bias Effect Estimates in GWAS) can be directly applied to this pleiotropic study.

Reviewer #3 (Remarks to the Author):

The authors submitted a rebuttal for this paper, but in my opinion, they failed to address most of my concerns. For most of the major points, the authors provided incomplete, biased, or incorrect arguments, and little new analysis or investigation was performed in the revision to address these points. Specifically,

1. The authors insisted on reporting all the significant SNPs and did not even try to answer the question 'how many independent associations' there were. I don't think this is acceptable, as this is essential in almost any GWAS report, and the number of SNPs can simply be misleading.
2. The 607-SNP replication part is still not very clear to me. So the 607 SNPs have $p < 1e-4$ in both eMERGE and UKB? How many do we expect under the null? The text says 134,309 SNPs had $p < 1e-4$ in eMERGE, but the number doesn't match Figure 2. All the mess is related to the poor locus definition without knowing where the independent hits are.
3. Without considering independent associations, Figure 2 is not informative, but the authors 'still think' it is enough to simply look at the SNPs. I don't understand why. Also, related to the replication thing above, most SNPs 137k + 23k were only significant in eMERGE or UKB PheWAS, very poor replication?
4. Re univariate v.s. multivariate tests, the authors claimed 'readers can get whichever comparisons' they want from Figure 2. But no, obviously multivariate tests can discover much fewer SNPs, why? The readers do not only want the numbers but rather want to understand what causes the difference.
5. The author claim that they 'leave the number of independent associations open-ended', because COJO or SOJO have their problems? All methods have their problems, but they provide useful results for inference. I don't think it's acceptable to refuse such an analysis. The authors neglected the colocalization analysis comment.
6. In the reply, the authors seem to say the five loci are novel. Are they? I asked at the beginning of my original report, but they seem to have neglected the point.
7. Re Figure 4, LD has to be considered, otherwise I don't think this analysis is correct. A strong SNP with many others in high LD will simply be over-weighted in the proportion calculation. Colocalization analysis can help here, but the authors neglected it.
8. The authors seem to have tested LD score regression in estimating genetic correlations, but the results were not shown (even if not powerful enough). If using individual-level data, power should be much stronger. There actually exists a new summary stats based method more powerful than LD score regression (Nat Genet 52, 859-864). Overall, this is a major point that can be better answered.

9. Re other summary-level multivariate methods, the authors were simply wrong about them. They don't consider sample overlap as a nuisance. MultiABEL is summary stats based MANOVA, so in perfectly overlapped samples for multiple traits, it should give the same answer as individual-level data method such as MultiPhen.

10. Why is it not appropriate to interpret multivariate genetic effects? I believe there is always some interpretation, instead of saying that those effects are meaningless. The p-value comparison with the univariate method is strange; why do they have the same alternative hypothesis? In the multivariate alternative hypothesis, we don't specify which univariate effect is non-zero.

REVIEWER COMMENTS

Reviewer #1 (Remarks to the Author):

Zhang et al. revised the manuscript in response to the reviewer's comments and it appears to have been improved substantially. The reviewer has a few comments on point #1. Since there are two types of effect direction, same and opposite, at the loci showing significant association with distinct (but similar combination of) phenotypes (or disease traits), we had better be careful about the interpretation. Even though the authors performed a statistical test of pleiotropy, is it still possible that not a single gene variant but multiple alleles at the gene of interest may be present, with the individual alleles exerting the opposite direction of effect on multiple phenotypes? Alternatively, is it possible that multiple variants (SNPs) of adjacent but distinct genes may be located on the same haplotype in the associated region, ie, in strong LD, which could explain some of the finding of antagonistic pleiotropy? If so, such an observation should be not a pleiotropy but a composite of separate SNP-disease associations. Please discuss these possibilities.

Thank you for the comment. We performed our analysis on single genetic variants (SNPs) per the design of the analytical model, so the reported synergistic and antagonistic pleiotropy results were observed for a

single variant that presented in each of the biobanks, instead of loci - which represents multiple genetic variants in each nearby region. However, only reporting SNP-level results may not be appropriate (as was suggested by reviewer #3), and so we reported both loci-level results and all SNP-level results in the manuscript. In addition, we performed additional colocalization analyses (see response for reviewer #3 comment #5), and the results are suggesting that our reported loci had the same underlying casual signals across circulatory and nervous system traits. As to the reviewer’s second point, it is certainly possible that one tag SNP (in the genotyping array) represented two different underlying casual SNPs as the reviewer has suggested, however, it is challenging to test this hypothesis given the available genotype data in the biobanks. This scenario that the reviewer mentioned is also discussed as “spurious pleiotropy” in Solovieff *et al.* Nature Genetics 2013 paper. The authors stated that tagSNPs in HLA region might tag multiple genes and it is particularly challenging to distinguish them between biological and spurious pleiotropy. It is a great point and we’ve added it to the discussion (Lines 392-396). Thank you for making this very important suggestion as we do need to be careful that we do not mis-interpret the results.

Reviewer #2 (Remarks to the Author):

In line with Q1 in the previous report, I have a further question for the line 198 – 199, and for the authors’ response “... we chose not to do a comprehensive pairwise analyses owing to the immense multiple comparison burden as well as because it still wouldn’t help conclude with certainty if one or both traits are associated with the SNP”.

It is still not clear if the significance of the 52 SNPs in eMERGE and 59 SNPs in UKBB are entirely due to pleiotropy between circulatory and nervous system disorders or due to something else (e.g. pleiotropy within circulatory (or nervous) system disorders) (see Figure 5). I am not quite sure that the threshold used in the formal test of pleiotropy can be justified well. I would suggest the authors should do comprehensive pairwise analyses with a proper multiple test correction and should discuss the difference in their results if there is significant difference.

Thanks for the comments. In lines 195-197, we stated that the pleiotropic SNPs that we reported are from both disease categories (at least one nervous system disorder and one circulatory disease), which is the main question that we are addressing in this manuscript. We hope it is clear to the reader.

We appreciate the reviewer’s comment on pairwise analyses among traits and thus we performed comprehensive pairwise MultiPhen analyses on the five independent loci where we tested one nervous system phenotype and one cardio phenotype. We present our comparison results in the below table. Not surprisingly, we do see differences in the p-value, which is due to different phenotypes included in each of the models, where each phenotype will have different numbers of cases and controls. The joint model includes all phenotypes whereas the pairwise analyses only include one nervous and one circulatory phenotype. From our results, we see that each pairwise comparison is statistically significant even after Bonferroni correction (the corrected p-value threshold is 1.8×10^{-4} based on 27 pairwise tests). We also see that the p-value for the joint MultiPhen model that contains all phenotypes remains to be statistically significant for our reported pleiotropy (also after Bonferroni correction).

Chr	rsID	Cardio_pheno	Nervous_pheno	Pairwise MultiPhen P-value	Joint Model MultiPhen P-value
19	19_45396219	Atherosclerotic_heart_disease	Dementia	1.38144E-29	
19	19_45396219	Atherosclerotic_heart_disease	Delirium	4.35102E-11	1.92E-26

19	19_45396219	Atherosclerotic heart disease	Alzheimer's disease	4.21941E-34	
4	4_81184341	Essential (primary) hypertension	Severe depressive episode with psychotic symptoms	3.33734E-47	4.92517E-24
9	9_22114495	Acute myocardial infarction	Major depressive affective disorder, recurrent episode, moderate	2.63552E-08	
9	9_22114495	Intermediate coronary syndrome	Major depressive affective disorder, recurrent episode, moderate	1.70327E-10	
9	9_22114495	Old myocardial infarction	Major depressive affective disorder, recurrent episode, moderate	1.19892E-12	
9	9_22114495	Angina pectoris	Major depressive affective disorder, recurrent episode, moderate	4.03234E-09	
9	9_22114495	Coronary atherosclerosis	Major depressive affective disorder, recurrent episode, moderate	2.78234E-07	
9	9_22114495	Coronary atherosclerosis of unspecified type of vessel, native or graft	Major depressive affective disorder, recurrent episode, moderate	4.36372E-20	
9	9_22114495	Coronary atherosclerosis of native coronary artery	Major depressive affective disorder, recurrent episode, moderate	1.02088E-16	
9	9_22114495	Coronary atherosclerosis of autologous vein bypass graft	Major depressive affective disorder, recurrent episode, moderate	1.97947E-11	
9	9_22114495	Coronary atherosclerosis of unspecified bypass graft	Major depressive affective disorder, recurrent episode, moderate	4.15643E-10	
9	9_22114495	Other specified forms of chronic ischemic heart disease	Major depressive affective disorder, recurrent episode, moderate	7.17496E-08	3.3262E-57
6	rs9273532	Hypertensive chronic kidney disease	Paralysis agitans	3.57828E-06	
6	rs9273532	Chronic pulmonary heart diseases	Paralysis agitans	2.40609E-06	
6	rs9273532	Conduction disorder	Paralysis agitans	6.51109E-05	
6	rs9273532	Atherosclerosis of native arteries of the extremities, unspecified	Paralysis agitans	1.28609E-09	
6	rs9273532	Atherosclerosis of native arteries of the extremities with intermittent claudication	Paralysis agitans	1.08253E-09	
6	rs9273532	Peripheral vascular disease	Paralysis agitans	3.46966E-11	
6	rs9273532	Hypertensive chronic kidney disease	Multiple Sclerosis	7.66428E-09	
6	rs9273532	Chronic pulmonary heart diseases	Multiple Sclerosis	5.30402E-09	
6	rs9273532	Conduction disorder	Multiple Sclerosis	1.32534E-07	
6	rs9273532	Atherosclerosis of native arteries of the extremities, unspecified	Multiple Sclerosis	1.97062E-12	
6	rs9273532	Atherosclerosis of native arteries of the extremities with intermittent claudication	Multiple Sclerosis	1.55135E-12	
6	rs9273532	Peripheral vascular disease	Multiple Sclerosis	6.05961E-14	5.47E-08
6	6_32765182	Pulmonary embolism and infarction	Multiple Sclerosis	5.1124E-13	1.15364E-21

For Q5, the authors can transform the ordinal or class variable to n x m matrix with 0 and 1 where n is the same size and m is the number of levels of the variable. Or, the phenotypes can be pre-adjusted for the variable before the main analysis. I don't see why this is not possible to test.

The reviewer is commenting on treating the centre as a covariate for UK Biobank data that we tried in the first round of revision. Thanks for bringing this up. The centre variable has 22 unique values in the UK Biobank. We tried to encode it as dummy variable and performed sequential multivariate analysis to

address the reviewer's concern. However, the software failed the test by giving the error "system is computationally singular". We then tried to adjusted this variable by treating it as a continuous variable knowing that the variable should be ordinal. The results of this analysis show that our discovery results remain the same when adjusting for centre as a quantitative variable in the UK Biobank. Thus, we are confident with the results as they are. But we appreciate the opportunity to explain this further.

Finally, I wonder if the relationship between circulatory and nervous system disorders is mediated via genes involved in obesity. It would be useful to see how the results will be changed when the phenotypes should be adjusted for BMI. I am not sure if the issue raised in the paper (Adjusting for Heritable Covariates Can Bias Effect Estimates in GWAS) can be directly applied to this pleiotropic study.

We evaluated BMI for the five loci as suggested by the reviewer. The results are shown in the below table. We observe that three out of five loci (rs157582, rs9273532 and rs10811656) are associated with both circulatory and nervous disease. There are two loci (rs16998073 and rs7767167) that are associated with only one disease category. There are several possible reasons behind this. First, the sample size is reduced as the number of people with BMI measurements is a subset of the original dataset. For eMERGE, it reduced from 43,015 to 35,696. For UK Biobank, it reduced from 295,423 to 294,342. Second, it is possible that the evidence of pleiotropy for these two loci are in fact affected by BMI (e.g. mediated by obesity as the reviewer is suggesting). We agree that it is a reasonable hypothesis that BMI/obesity could play a role in the interrelationship between circulatory and nervous diseases. Future work on the full evaluation of obesity and related traits such as diabetes would be very useful.

SNP	Not adjust for BMI		Adjust for BMI	
	Circulatory Trait	Nervous Trait	Circulatory Trait	Nervous Trait
rs157582	Atherosclerotic heart disease	Dementia Delirium Alzheimer's disease	Atherosclerotic heart disease Other specified cerebrovascular diseases	Dementia Delirium Alzheimer's disease
rs16998073	Essential (primary) hypertension	Severe depressive episode with psychotic symptoms	Essential (primary) hypertension Atrial fibrillation and flutter	
rs9273532	Hypertensive chronic kidney disease Other chronic pulmonary heart diseases Conduction disorder Atherosclerosis of native arteries of the extremities Peripheral vascular disease	Paralysis agitans Multiple Sclerosis	Hypertensive chronic kidney disease Cerebral artery occlusion Atherosclerosis of native arteries of the extremities Atherosclerosis of aorta	Multiple Sclerosis
rs7767167	Other pulmonary embolism and infarction	Multiple Sclerosis		Multiple Sclerosis

rs10811656	Acute myocardial infarction, Subendocardial infarction, initial episode of care Intermediate coronary syndrome Old myocardial infarction Other and unspecified angina pectoris Coronary atherosclerosis Coronary atherosclerosis of unspecified type of vessel, native or graft Coronary atherosclerosis of native coronary artery Coronary atherosclerosis of autologous vein bypass graft Coronary atherosclerosis of unspecified bypass graft Other specified forms of chronic ischemic heart disease Chronic ischemic heart disease, unspecified Occlusion and stenosis of carotid artery without mention of cerebral infarction Occlusion and stenosis of multiple and bilateral precerebral arteries without mention of cerebral infarction Atherosclerosis of native arteries of the extremities, unspecified Atherosclerosis of native arteries of the extremities with intermittent claudication Abdominal aneurysm without mention of rupture Peripheral vascular disease, unspecified	Major depressive affective disorder, recurrent episode	Acute myocardial infarction, Subendocardial infarction, initial episode of care Intermediate coronary syndrome Old myocardial infarction Other and unspecified angina pectoris Coronary atherosclerosis Coronary atherosclerosis of unspecified type of vessel, native or graft Coronary atherosclerosis of native coronary artery Coronary atherosclerosis of autologous vein bypass graft Coronary atherosclerosis of unspecified bypass graft Chronic ischemic heart disease, unspecified Occlusion and stenosis of carotid artery without mention of cerebral infarction Occlusion and stenosis of multiple and bilateral precerebral arteries without mention of cerebral infarction Atherosclerosis of native arteries of the extremities, unspecified Atherosclerosis of native arteries of the extremities with intermittent claudication Abdominal aneurysm without mention of rupture Peripheral vascular disease, unspecified	Major depressive affective disorder, recurrent episode
------------	---	---	---	--

Reviewer #3 (Remarks to the Author):

The authors submitted a rebuttal for this paper, but in my opinion, they failed to address most of my concerns. For most of the major points, the authors provided incomplete, biased, or incorrect arguments, and little new analysis or investigation was performed in the revision to address these points. Specifically,

1. The authors insisted on reporting all the significant SNPs and did not even try to answer the question 'how many independent associations' there were. I don't think this is acceptable, as this is essential in almost any GWAS report, and the number of SNPs can simply be misleading.

We understand that this is a very important issue and have reported our results specified as loci rather than SNPs in the main text as the reviewer has suggested. We think that it is important from the point of reproducibility of research to also include the SNP results in the tables so that other people can reproduce our study rather than simply look at the loci that we identify as statistically significant. However, we greatly appreciate this point and do not want to overstate our results. Thus, we focus primarily on loci rather than SNP.

2. The 607-SNP replication part is still not very clear to me. So the 607 SNPs have $p < 1e-4$ in both eMERGE and UKB? How many do we expect under the null? The text says 134,309 SNPs had $p < 1e-4$ in eMERGE, but the number doesn't match Figure 2. All the mess is related to the poor locus definition without knowing where the independent hits are.

First, we apologize that this reviewer thinks this section is “**all the mess**”. The reviewer is correct that in the previous version of our manuscript, 607 SNPs have $p < 1e-4$ in both eMERGE and UKBB using both methods. The number 134,363 SNPs matches Figure 2 (adding all SNPs – which represents the total number of variants that were tested in both eMERGE and the UKBB). To clarify this point and reduce “the mess”, we added a sentence at the end of the legend for Figure 2. We also replaced Figure 2 with the number of independent loci as the reviewer suggested.

3. Without considering independent associations, Figure 2 is not informative, but the authors 'still think' it is enough to simply look at the SNPs. I don't understand why. Also, related to the replication thing above, most SNPs 137k + 23k were only significant in eMERGE or UKB PheWAS, very poor replication?

We replaced Figure 2 with independent loci. The eMERGE dataset has an overall larger number of SNPs with p-values larger than $1e-8$ comparing to UKBB – SNPs under the red line in Figure 1. UKBB has higher sample size, thus might reduce false positives that could be potentially identified by a smaller dataset such as eMERGE. We are using a p-value of $1e-4$ to select as many SNPs as possible to conduct the replication analysis. When applying Bonferroni significance threshold (Fig.S2B), we see that among 451 ($354+76+13+8=451$) independent loci from eMERGE, UKBB identified 438 of them ($451-13=438$).

4. Re univariate v.s. multivariate tests, the authors claimed 'readers can get whichever comparisons' they want from Figure 2. But no, obviously multivariate tests can discover much fewer SNPs, why? The readers do not only want the numbers but rather want to understand what causes the difference.

Thanks for your comments. Figure 2 was based on an exploratory p-value threshold. When looking at Bonferroni significant plot (Fig. S2B), for eMERGE, PheWAS identified 451 loci ($354+76+13+8$) whereas MultiPhen identified 82 loci ($76+3+3$); for UKBB, PheWAS identified 1064 loci ($408+354+215+76+8+3$) whereas MultiPhen identified 842 loci ($408+354+76+3+1$). MultiPhen discovered fewer SNPs for discovery and replication analyses. According to MultiPhen paper (O'Reilly *et al*), univariate method is slightly more powerful when the genetic effects are consistent with the correlation between the two phenotypes, e.g. when a variant has the same effect on two highly correlated phenotype, or only affects one of the two uncorrelated phenotypes. It is possible that the scenario when univariate methods are more powerful exists more in our datasets, especially for variants that only affects one of tested phenotype. MultiPhen, on the other hand, has more power when variant affects more less correlated phenotype, or affects multiple phenotypes instead of one. We've added the explanation in the Results (Lines 169-172).

5. The author claim that they 'leave the number of independent associations open-ended', because COJO or SOJO have their problems? All methods have their problems, but they provide useful results for inference. I don't think it's acceptable to refuse such an analysis. The authors neglected the colocalization analysis comment.

We defined the independent loci using LD pruning on the genotype data and reported them in the manuscript. COJO or SOJO can be applied when individual data are not available. We performed extensive colocalization analyses as the reviewer suggested.

We performed colocalization analyses on three loci. The other two loci are located near the HLA region, and we decide not to perform colocalization because of the complexity of the region. Since the discovery of pleiotropy is within the dataset, we performed the colocalization analysis for selected pairs of traits within each dataset. We acknowledge that the posterior probabilities can be biased when we perform these analyses between two eMERGE or two UKBB signals since there will be sample overlap, and we decided to put these results in the response to the reviewers' comments but not to include them in the main manuscript. Among the three loci we tested, we do observe colocalization implications ($PP4 / (PP4 + PP3) \geq 0.8$) from all three loci for at least one circulatory and one nervous trait (results shown below). We also updated the locuszoom plots in the manuscript.

Here are the locuszoom plots with colocalization results included:

SNP rs16998073:

SNP rs157582:

SNP rs10811656:

6. In the reply, the authors seem to say the five loci are novel. Are they? I asked at the beginning of my original report, but they seem to have neglected the point.

We have previously addressed the issue in the first round of revision accordingly in the manuscript (see previous report Reviewer #1 comment #2). Here is the response again for clarification:

“Thank you for this observation. We discussed the novelty of each region in the Results section and Discussion section in detail by reviewing previous literature and the NHGRI/EBI GWAS catalog. For the ApoE region, as we stated in the manuscript (Results section), it has been shown to be associated with Alzheimer’s disease and cardiovascular disease risk factors such as HDL, LDL according to the GWAS catalog. We also reported a SNP in ApoE that was known for coronary artery disease. We agree with the reviewer that this region was a validation and state the finding as a “positive control” in Lines 235-236. For the other novel regions, we put a more detailed discussion of the novelty for each region in the main manuscript (Lines 264, 275, 382). One additional aspect of novelty in this study is that most of previous findings have been conducted in independent studies (e.g. GWAS studies), but our work provides evidence that using a unified analytical framework to formally characterize pleiotropy can be a fruitful endeavor.”

Thus, out of the five novel loci, some of the associations with one of the traits in the pleiotropy models may have been already known (thus a positive control or validation); however the other trait is novel and/or the pleiotropy evidence is novel. We hope that this better clarifies this point.

7. Re Figure 4, LD has to be considered, otherwise I don't think this analysis is correct. A strong SNP with many others in high LD will simply be over-weighted in the proportion calculation. Colocalization analysis can help here, but the authors neglected it.

Thank you so much for the comment. We took the comment of focusing on the independent loci and removed the proportion from the Figure. We have generated a new Figure 4. Please see colocalization results in comment #5.

8. The authors seem to have tested LD score regression in estimating genetic correlations, but the results were not shown (even if not powerful enough). If using individual-level data, power should be much stronger. There actually exists a new summary stats based method more powerful than LD score regression (Nat Genet 52, 859-864). Overall, this is a major point that can be better answered.

As we stated in the first round of revision, “However, the LDSC package suggested that the sample size for eMERGE is too small such that it cannot give us a reliable estimation on the genetic correlation.” This is what was reported when we ran the software. The exact error message is “WARNING: One of the h^2 's was out of bounds. This usually indicates a data-munging error or that h^2 or N is low.” The estimation for genetic correlation between all pairs of traits is NA.

LDSC requires larger sample size to make it work and the eMERGE sample size is not large enough for LDSC. The summary stat based method that suggested by the reviewer performs better than LDSC by doing a great job in taking LD information, but we would have the same issue due to the small sample size in eMERGE. As we stated in the first round of revision, such summary-stat based method can be used for large sample size including large-scale meta-analysis. And such methods hold great promises for those analyses. However, it is not appropriate for the sample size of our data.

9. Re other summary-level multivariate methods, the authors were simply wrong about them. They don't consider sample overlap as a nuisance. MultiABEL is summary stats based MANOVA, so in perfectly overlapped samples for multiple traits, it should give the same answer as individual-level data method such as MultiPhen.

Thank you for the suggestion. We appreciate the reviewer's suggestion on the summary-based method. We read the MultiABEL paper published on bioRxiv in 2019. From our understanding, the phenotype assumption of MultiABEL is normally-distributed. Here is a quote from the paper “we assume the phenotypes are standardized to have a mean zero and variance of one”, and two major statistical models CCA and MANOVA used in MultiABEL assume normally-distributed phenotype. The phenotype in our study is case control – binary phenotypes. O'Reilly evaluated in the MultiPhen paper (O'Reilly et al 2012) about the application of CCA and MANOVA on binary phenotypes, both methods demonstrated inflated type-1 error rates, while MultiPhen does not have such inflation. Thus, we do not believe that MultiABEL is an appropriate tool for the analyses in this manuscript.

We would also like to point out that when given the choice between using summary-based methods or individual-level data, it is certainly a better approach to use the data available. Thus, while MultiABEL may not be an appropriate method for this analysis, we also did not seek out other summary-based methods because we are in the fortunate position to have individual-level data for this project.

10. Why is it not appropriate to interpret multivariate genetic effects? I believe there is always some interpretation, instead of saying that those effects are meaningless. The p-value comparison with the univariate method is strange; why do they have the same alternative hypothesis? In the multivariate alternative hypothesis, we don't specify which univariate effect is non-zero.

We apologize that we have created some confusion here. First, the p-value from a multivariate model refers to the statistical significance of the joint modeling of all phenotypes in the model; unfortunately, there is no corresponding beta for the joint p-value. If we want to interpret a coefficient for each phenotype in a multivariate model, each coefficient can be interpreted when we assume other predictor variables are held constant. In other words, the coefficient can change when the phenotypes (as predictors) in multivariate model change. The null hypothesis of MultiPhen is to test if " $\beta_1=\beta_2=\dots=\beta_k=0$ ", and the test is designed for testing this null hypothesis only. The rejection of null hypothesis is met when at least one of the betas does not equal zero. That is the interpretation we should draw from the model, which is based on the p-value. The lack of the ability to pinpoint the specific associated traits is also the motivation of the development of the "pleio" method. Here is a quote from the paper (Biostatistics 2019, 20, 1, pp111-128): "most current multivariate methods to evaluate pleiotropy test the null hypothesis that none of the traits are associated with a variant; departures from the null could be driven by just one associated trait". In this manuscript, we simply used a univariate test and a multivariate test as a pre-selection technique to select potential SNPs (with liberal p-value) and subsequently performed a formal test of pleiotropy using pleio to pinpoint the specific associated traits.

Second, the reviewer is right that in the multivariate alternative hypothesis, we do not specify which univariate effect is non-zero. In order to make univariate and multivariate methods comparable, in our study, we used the minimum univariate p-value across all tested phenotypes while we do not assume which effect is non-zero – this is the standard way to compare univariate and multivariate methods (e.g. O'Reilly et al 2012). Again, both univariate and multivariate are two ways to pre-select variants that go into "pleio" test for a formal test of pleiotropy – as the heavy computational burden for "pleio" test. We hope that this adds clarity to this issue.

REVIEWER COMMENTS

Reviewer #1 (Remarks to the Author):

The authors have responded to the reviewer's comments and have revised the manuscript almost satisfactorily.

Reviewer #2 (Remarks to the Author):

The authors have addressed most of my concerns, and the manuscript has been significantly improved. I have no further comments.

Reviewer #3 (Remarks to the Author):

The authors submitted another rebuttal for the paper, but in my opinion, the revision is not satisfactory. It appears to me that the paper still lacks a logically clear statistical inference procedure, which affects the scientific reasoning in the genetic analysis too. Following my previous comments, I list some points below.

1. Before the other points related to SNPs and loci definition, first of all, the authors refused to use a user-friendly tool such as GCTA-COJO to identify independent significant SNPs. LD pruning was applied, but with PLINK parameters "--indep-pairwise 100 5 0.8" -- Why 0.8? Any justifications? 0.8 seems WAY TOO HIGH as an r^2 threshold for pairwise LD. In order to claim independent associations, better go for 0.01 (corresponding to a correlation between -0.1 and 0.1), unless a higher threshold can be justified.

2. I'm so bothered why "SNPs with $p \leq 1 \times 10^{-4}$ across all tested 138 phenotypes in eMERGE" were passed onto UKBB PheWAS analysis for replication. Did the authors ever consider FDR in discovery? $642,122 \times 138 \times 1e-4$ gives 105,461 false positives under the null. So most of the 134,363 to be replicated are expected to be false. One cannot simply lower the GWAS significance threshold this much due to the severe multiplicity problem in the genome scan, especially here for so many traits. So it appears Fig. S2 with Bonferroni correction is more correct than Fig. 2. However, due to point 1 above, the number of independent associations is incorrect or not justified.

3. Again on the FDR point, 134,363 SNPs were analyzed in the UKBB replication across 102 traits. We expect $134,363 \times 1e-4 \times 102 = 1,371$ false positives under the null. So comparing to the number 1,414, how can these be justified as "replicated"?

4. In the reply to my previous comments, the authors stated "UKBB has higher sample size, thus might reduce false positives that could be potentially identified by a smaller dataset such as eMERGE." Why would a large sample size reduce false positives? I don't think so. The p-value distribution under the null

is always uniform, regardless of sample size.

5. In the reply to my previous comments, the authors conducted some colocalization analysis using coloc. Regardless of whether the posteriors are biased or not, aren't these results directly supportive of results such as Figure 4? Why not include them in the supplements as then we know the blue and red bars in Figure 4 are likely caused by the same variants?

6. I kind of understand the logic that the authors performed univariate and multivariate GWAS to start with, then "subsequently performed a formal test of pleiotropy using pleio to pinpoint the specific associated traits." If so, I don't feel the current results clearly show the point of the multivariate/pleiotropic analysis -- Which traits were pinpointed? By what criteria of the "Pleio" method? Are the pinpointed traits the same in discovery and replication? For the same pinpointed trait, are the effects going the same direction in discovery and replication, at least when looking at univariate betas? We all know that a p-value replication is not really a replication -- the genotype-phenotype maps need to be consistent in discovery and replication in order to claim "replicated".

7. The authors didn't agree with my comment on the discussion about summary-based methods. So I see e.g. lines 316-319, 327-329 are still there. Let me try to explain my point here again, as I do think most of these statements are incorrect or not justified. I don't ask the authors to do the analysis using a summary-based method instead, as it would be unnecessary. However, if after reading my explanation below, the authors still insist on this piece of discussion, I would require a simulation to justify, as I can only see advantages rather than disadvantages of summary-based methods compared to individual-level data methods.

The bias of CCA and MANOVA for binary phenotypes does not directly apply to summary-based methods. The methods using GWAS summary statistics for binary traits are not analyzing the binary phenotypes themselves but rather the underlying liability. This is because a logistic regression was used in the GWAS analysis, not an ordinary linear regression. This not only applies to multi-trait analysis methods but also genetic correlation analysis methods such as LDSC. This is actually an advantage of summary-based methods over individual-level data methods. Technically, a summary-based method DOES NOT KNOW the distribution of the phenotype, as we DON'T KNOW (by looking at the GWAS summary statistics) whether the betas are linear regression slopes or logistic regression log odds ratios. The underlying liability has a logistic distribution, which is nicely smooth and symmetric, thus fine for many standard analyses.

8. Later in the Discussion, the authors were talking about how slow it is to run the "formal" test of pleiotropy, then stated, "Future development of more computationally efficient methods that use individual-level data, rather than summary statistics, would greatly facilitate the detection of pleiotropy." Is this really what was intended to be said? Isn't the current "Pleio" method using individual-level data? Shouldn't a summary-level method be much faster? I believe this is certainly doable. By looking at e.g. <https://github.com/xiashen/MultiABEL/>, it seems rather straightforward to

perform summary-level multi-trait analysis on different combinations of phenotypes, though software as such has not implemented the exact sequential procedure as "Pleio" introduced.

Reviewer #1 (Remarks to the Author):

The authors have responded to the reviewer's comments and have revised the manuscript almost satisfactorily.

Reviewer #2 (Remarks to the Author):

The authors have addressed most of my concerns, and the manuscript has been significantly improved. I have no further comments.

Reviewer #2 comment on Reviewer #3 report:

#1. Although I don't think this is a critical point, the authors can do a sensitivity analysis (varying the r^2 threshold for pairwise LD).

We agree with the reviewer's suggestion and therefore we evaluated another r^2 threshold. Here, we used an r^2 of 0.1 as the threshold and modified the number of loci in the manuscript (See Method PheWAS section). We also updated the Figure 2, Supplementary Figure S2B and relevant content based on the above LD threshold. The major findings do not change based on the new LD threshold, however, because this more stringent threshold reduced the number of independent loci, we have reported these results in the manuscript. The updated methods are reported on lines 518-524.

For the reviewers, here are the two sets of results:

R^2 threshold	$R^2 = 0.1$	$R^2 = 0.8$
Number of SNPs in eMERGE	145,131	145,131
Number of independent loci in eMERGE	11,822	39,521
Number of SNPs in UKBB	134,363	134,363
Number of independent loci in UKBB	10,472	35,352

#2 and #3. I agree that the authors should clarify why they used an arbitrary threshold with $p \leq 1 \times 10^{-4}$. However, as shown in Fig S1., the formal test of pleiotropy was done with the standard genome-wide significant threshold, i.e. $p < 1 \times 10^{-8}$) in the final step.

Thank you for your feedback. We explained the use of the $1e-4$ in lines 112-118. Here is what was stated in the text "A formal systematic replication analyses was conducted in UKBB on 134,363 genetic variants that had an exploratory p-value significance of $p \leq 1 < 10^{-4}$ from analyses in eMERGE dataset (and passed QC in the UKBB dataset). The use of an exploratory p-value threshold such as 1×10^{-4} enables exploration of genetic variants beyond the most significant signals at a genome-wide significance threshold. Other studies have employed this strategy and it can be beneficial to identify variants

that may not meet genome-wide significance in one dataset but otherwise be potentially informative¹².” We also cite our previously manuscript, Verma, A. et al. 2018, where we used this same exploratory p-value. While we chose this threshold to be inclusive of potentially informative signals to take forward to our formal test for pleiotropy, we stuck to the genome-wide significance threshold for the *pleio* analyses to avoid any p-hacking.

#4. The reviewer’s point is correct. Sample size determines the power, so it should be ‘... reduce false negatives ...’. This can be amended by the author.

Thank you for the comments, as explained in our last round of the revision, we suggested that the use of UKBB as a replication dataset could help with identifying true positive signals. We agree with reviewer’s comment on this and our use of the phrase might be misleading. We should have said reduce false negatives. Since this was only mentioned in our last round of the response to reviewer’s comments, we did not edit any content in the manuscript. There is no discussion of this in the manuscript.

#5. It seems feasible for the authors to incorporate this comment.

Thanks for the comment. We have incorporated this part of results into the supplementary file (See *Colocalization Analyses* section in the Supplement lines 2-14).

#6. I agree with this comment and the authors should address this question clearly in their Method section.

The traits in the ‘pleio’ method are identified by using a sequential multivariate model, that iteratively tests combinations of traits to see which are significant at the 1e-8 p-value threshold (We have explained how the pleio method works in the Methods section). The results of the ‘pleio’ approach will tell us which traits are associated with the genetic variant, but like other multivariate frameworks it does not provide a provide biological meaningful multivariate beta (please see our elaborated explanation at the last point of last round of revision). The genetic effect (betas) reported in the manuscript are obtained from the PheWAS (are provided in summary form in Table S3 and in full details in Table S7). We added more explanation of this in the methods section (Lines 560-564). As we stated in the paper, each method has its own pros and cons, and we utilize their advantages as complementary approaches to improve our ability to make genetic discoveries.

#7 and #8. These points are for Discussion section, which wouldn’t affect the main finding and conclusion. One note is that GWAS summary-based method is faster, but the accuracy or precision is reported to be lower, compared to individual-level data methods. The authors can read some previous relevant studies (e.g. see below) and use them as a reference.

Estimation of Genetic Correlation via Linkage Disequilibrium Score Regression and

Genomic Restricted Maximum Likelihood. Am J Hum Genet. 2018 Jun 7; 102(6): 1185–1194.

Thank you for your feedback. We have added this citation in the discussion for the summary-level methods part (Line 343-346). Aside from the bias in the estimates and issues with model assumptions (e.g., they usually require very large N), another drawback of summary-based methods is the requirement of an external LD reference panel which might not necessarily comport with the LD distribution of the considered dataset and can pose issues in the presence of admixture.

Reviewer #3 (Remarks to the Author):

The authors submitted another rebuttal for the paper, but in my opinion, the revision is not satisfactory. It appears to me that the paper still lacks a logically clear statistical inference procedure, which affects the scientific reasoning in the genetic analysis too. Following my previous comments, I list some points below.

1. Before the other points related to SNPs and loci definition, first of all, the authors refused to use a user-friendly tool such as GCTA-COJO to identify independent significant SNPs. LD pruning was applied, but with PLINK parameters "--indep-pairwise 100 5 0.8" -- Why 0.8? Any justifications? 0.8 seems WAY TOO HIGH as an r^2 threshold for pairwise LD. In order to claim independent associations, better go for 0.01 (corresponding to a correlation between -0.1 and 0.1), unless a higher threshold can be justified.

As described above, we have done a sensitivity analysis whereby we used the lower proposed r^2 value of 0.1. We agree with the reviewer's suggestion and therefore we evaluated another r^2 threshold. Here, we used an r^2 of 0.1 as the threshold and modified the number of loci in the manuscript (See Method PheWAS section). We also updated the Figure 2, Supplementary Figure S2B and relevant content based on the above LD threshold. The major findings do not change based on the new LD threshold, however, because this more stringent threshold reduced the number of independent loci, we have reported these results in the manuscript. The updated methods are reported on lines 518-524.

For the reviewers, here are the two sets of results:

R^2 threshold	$R^2 = 0.1$	$R^2 = 0.8$
Number of SNPs in eMERGE	145,131	145,131
Number of independent loci in eMERGE	11,822	39,521
Number of SNPs in UKBB	134,363	134,363
Number of independent loci in UKBB	10,472	35,352

Overall, the conclusions of the manuscript do not change significantly, however, this more stringent threshold reduces the number of independent loci reported.

2. I'm so bothered why "SNPs with $p \leq 1 \times 10^{-4}$ across all tested 138 phenotypes in eMERGE" were passed onto UKBB PheWAS analysis for replication. Did the authors ever consider FDR in discovery? $642,122 \times 138 \times 1e-4$ gives 105,461 false positives under the null. So most of the 134,363 to be replicated are expected to be false. One cannot simply lower the GWAS significance threshold this much due to the severe multiplicity problem in the genome scan, especially here for so many traits. So it appears Fig. S2 with Bonferroni correction is more correct than Fig. 2. However, due to point 1 above, the number of independent associations is incorrect or not justified.

3. Again on the FDR point, 134,363 SNPs were analyzed in the UKBB replication across 102 traits. We expect $134,363 \times 1e-4 \times 102 = 1,371$ false positives under the null. So comparing to the number 1,414, how can these be justified as "replicated"?

Response to 2-3: Thank you for your feedback. We explained the use of the $1e-4$ in lines 114-116 of the revised manuscript. Here is what was stated in the text "A formal systematic replication analyses was conducted in UKBB on 134,363 genetic variants that had an exploratory p-value significance of $p \leq 1 \times 10^{-4}$ from analyses in eMERGE dataset (and passed QC in the UKBB dataset). The use of an exploratory p-value threshold enables studies of genetic variants beyond the most significant signals that may otherwise be potentially informative"¹². We also cite our previously manuscript, Verma, A. et al. 2018, where we used this same exploratory p-value. While we chose this threshold to be inclusive of potentially informative signals to take forward to our formal test for pleiotropy, we stuck to the genome-wide significance threshold for the *pleio* analyses to avoid any p-hacking.

4. In the reply to my previous comments, the authors stated "UKBB has higher sample size, thus might reduce false positives that could be potentially identified by a smaller dataset such as eMERGE." Why would a large sample size reduce false positives? I don't think so. The p-value distribution under the null is always uniform, regardless of sample size.

You are correct in that we mis-stated this in our previous response. We should have said reduce false negatives. Since this was only mentioned in our last round of the response to reviewer's comments, we did not edit any content in the manuscript. There is no discussion of this in the manuscript.

5. In the reply to my previous comments, the authors conducted some colocalization analysis using coloc. Regardless of whether the posteriors are biased or not, aren't these results directly supportive of results such as Figure 4? Why not include them in the supplements as then we know the blue and red bars in Figure 4 are likely caused by the same variants?

We have incorporated this part of results into the supplementary file (See *Colocalization Analyses* section).

6. I kind of understand the logic that the authors performed univariate and multivariate GWAS to start with, then "subsequently performed a formal test of pleiotropy using pleio to pinpoint the specific associated traits." If so, I don't feel the current results clearly show the point of the multivariate/pleiotropic analysis -- Which traits were pinpointed? By what criteria of the "Pleio" method? Are the pinpointed traits the same in discovery and replication? For the same pinpointed trait, are the effects going the same direction in discovery and replication, at least when looking at univariate betas? We all know that a p-value replication is not really a replication -- the genotype-phenotype maps need to be consistent in discovery and replication in order to claim "replicated".

The traits in the 'pleio' method are identified by using a sequential multivariate model, that iteratively tests combinations of traits to see which are significant at the $1e-8$ p-value threshold (We have explained how the pleio method works in the *Methods* section). The results of the 'pleio' approach will tell us which traits are associated with the genetic variant, but like other multivariate frameworks it does not provide a provide biological meaningful multivariate beta (please see our elaborated explanation at the last point of last round of revision). The genetic effect (betas) reported in the manuscript are obtained from the PheWAS (are provided in summary form in Table S3 and in full details in Table S7). We added more explanation of this in the methods section (Lines 560-564). As we stated in the paper, each method has its own pros and cons, and we utilize their advantages as complementary approaches to improve our ability to make genetic discoveries.

7. The authors didn't agree with my comment on the discussion about summary-based methods. So I see e.g. lines 316-319, 327-329 are still there. Let me try to explain my point here again, as I do think most of these statements are incorrect or not justified. I don't ask the authors to do the analysis using a summary-based method instead, as it would be unnecessary. However, if after reading my explanation below, the authors still insist on this piece of discussion, I would require a simulation to justify, as I can only see advantages rather than disadvantages of summary-based methods compared to individual-level data methods.

The bias of CCA and MANOVA for binary phenotypes does not directly apply to summary-based methods. The methods using GWAS summary statistics for binary traits are not analyzing the binary phenotypes themselves but rather the underlying liability. This is because a logistic regression was used in the GWAS analysis, not an ordinary linear regression. This not only applies to multi-trait analysis methods but also genetic correlation analysis methods such as LDSC. This is actually an advantage of summary-based methods over individual-level data methods. Technically, a summary-based method DOES NOT KNOW the distribution of the phenotype, as we DON'T KNOW (by looking at the GWAS summary statistics) whether the betas are linear regression slopes or logistic regression log odds ratios. The underlying liability has a logistic distribution, which is nicely smooth and symmetric, thus fine for many standard

analyses.

8. Later in the Discussion, the authors were talking about how slow it is to run the "formal" test of pleiotropy, then stated, "Future development of more computationally efficient methods that use individual-level data, rather than summary statistics, would greatly facilitate the detection of pleiotropy." Is this really what was intended to be said? Isn't the current "Pleio" method using individual-level data? Shouldn't a summary-level method be much faster? I believe this is certainly doable. By looking at e.g. <https://github.com/xiashen/MultiABEL/>, it seems rather straightforward to perform summary-level multi-trait analysis on different combinations of phenotypes, though software as such has not implemented the exact sequential procedure as "Pleio" introduced.

Thank you for your feedback. We have added this citation in the discussion for the summary-level methods part (Line 343-346). Aside from the bias in the estimates and issues with model assumptions (e.g., they usually require very large N), another drawback of summary-based methods is the requirement of an external LD reference panel which might not necessarily comport with the LD distribution of the considered dataset and can pose issues in the presence of admixture.

Reviewer #1 (Remarks to the Author):

The authors have responded to the reviewer's comments and have revised the manuscript almost satisfactorily.

Reviewer #2 (Remarks to the Author):

The authors have addressed most of my concerns, and the manuscript has been significantly improved. I have no further comments.

Reviewer #3 (Remarks to the Author):

The authors submitted another rebuttal for the paper, but in my opinion, the revision is not satisfactory. It appears to me that the paper still lacks a logically clear statistical inference procedure, which affects the scientific reasoning in the genetic analysis too. Following my previous comments, I list some points below.

1. Before the other points related to SNPs and loci definition, first of all, the authors refused to use a user-friendly tool such as GCTA-COJO to identify independent significant SNPs. LD pruning was applied, but with PLINK parameters "--indep-pairwise 100 5 0.8" -- Why 0.8? Any justifications? 0.8 seems WAY TOO HIGH as an r^2 threshold for pairwise LD. In order to claim independent associations, better go for 0.01 (corresponding to a correlation between -0.1 and 0.1), unless a higher threshold can be justified.

As described above, we have done a sensitivity analysis whereby we used the lower proposed r^2 value of 0.1. We agree with the reviewer's suggestion and therefore we evaluated another r^2 threshold. Here, we used an r^2 of 0.1 as the threshold and modified the number of loci in the manuscript (See Method PheWAS section). We also updated the Figure 2, Supplementary Figure S2B and relevant content based on the above LD threshold. The major findings do not change based on the new LD threshold, however, because this more stringent threshold reduced the number of independent loci, we have reported these results in the manuscript. The updated methods are reported on lines 518-524.

For the reviewers, here are the two sets of results:

R^2 threshold	$R^2 = 0.1$	$R^2 = 0.8$
Number of SNPs in eMERGE	145,131	145,131
Number of independent loci in eMERGE	11,822	39,521
Number of SNPs in UKBB	134,363	134,363
Number of independent loci in UKBB	10,472	35,352

Overall, the conclusions of the manuscript do not change significantly, however, this more stringent threshold reduces the number of independent loci reported.

2. I'm so bothered why "SNPs with $p \leq 1 \times 10^{-4}$ across all tested 138 phenotypes in eMERGE" were passed onto UKBB PheWAS analysis for replication. Did the authors ever consider FDR in discovery? $642,122 \times 138 \times 1e-4$ gives 105,461 false positives under the null. So most of the 134,363 to be replicated are expected to be false. One cannot simply lower the GWAS significance threshold this much due to the severe multiplicity problem in the genome scan, especially here for so many traits. So it appears Fig. S2 with Bonferroni correction is more correct than Fig. 2. However, due to point 1 above, the number of independent associations is incorrect or not justified.

3. Again on the FDR point, 134,363 SNPs were analyzed in the UKBB replication across 102 traits. We expect $134,363 \times 1e-4 \times 102 = 1,371$ false positives under the null. So comparing to the number 1,414, how can these be justified as "replicated"?

Response to 2-3: Thank you for your feedback. We explained the use of the $1e-4$ in lines 114-116 of the revised manuscript. Here is what was stated in the text "A formal systematic replication analyses was conducted in UKBB on 134,363 genetic variants that had an exploratory p-value significance of $p \leq 1 \times 10^{-4}$ from analyses in eMERGE dataset (and passed QC in the UKBB dataset). The use of an exploratory p-value threshold enables studies of genetic variants beyond the most significant signals that may otherwise be potentially informative"¹². We also cite our previously manuscript, Verma, A. et al. 2018, where we used this same exploratory p-value. While we chose this threshold to be inclusive of potentially informative signals to take forward to our formal test for pleiotropy, we stuck to the genome-wide significance threshold for the *pleio* analyses to avoid any p-hacking.

4. In the reply to my previous comments, the authors stated "UKBB has higher sample size, thus might reduce false positives that could be potentially identified by a smaller dataset such as eMERGE." Why would a large sample size reduce false positives? I don't think so. The p-value distribution under the null is always uniform, regardless of sample size.

You are correct in that we mis-stated this in our previous response. We should have said reduce false negatives. Since this was only mentioned in our last round of the response to reviewer's comments, we did not edit any content in the manuscript. There is no discussion of this in the manuscript.

5. In the reply to my previous comments, the authors conducted some colocalization analysis using coloc. Regardless of whether the posteriors are biased or not, aren't these results directly supportive of results such as Figure 4? Why not include them in the supplements as then we know the blue and red bars in Figure 4 are likely caused by the same variants?

We have incorporated this part of results into the supplementary file (See *Colocalization Analyses* section).

6. I kind of understand the logic that the authors performed univariate and multivariate GWAS to start with, then "subsequently performed a formal test of pleiotropy using pleio to pinpoint the specific associated traits." If so, I don't feel the current results clearly show the point of the multivariate/pleiotropic analysis -- Which traits were pinpointed? By what criteria of the "Pleio" method? Are the pinpointed traits the same in discovery and replication? For the same pinpointed trait, are the effects going the same direction in discovery and replication, at least when looking at univariate betas? We all know that a p-value replication is not really a replication -- the genotype-phenotype maps need to be consistent in discovery and replication in order to claim "replicated".

The traits in the 'pleio' method are identified by using a sequential multivariate model, that iteratively tests combinations of traits to see which are significant at the $1e-8$ p-value threshold (We have explained how the pleio method works in the Methods section). The results of the 'pleio' approach will tell us which traits are associated with the genetic variant, but like other multivariate frameworks it does not provide a provide biological meaningful multivariate beta (please see our elaborated explanation at the last point of last round of revision). The genetic effect (betas) reported in the manuscript are obtained from the PheWAS (are provided in summary form in Table S3 and in full details in Table S7). We added more explanation of this in the methods section (Lines 560-564). As we stated in the paper, each method has its own pros and cons, and we utilize their advantages as complementary approaches to improve our ability to make genetic discoveries.

7. The authors didn't agree with my comment on the discussion about summary-based methods. So I see e.g. lines 316-319, 327-329 are still there. Let me try to explain my point here again, as I do think most of these statements are incorrect or not justified. I don't ask the authors to do the analysis using a summary-based method instead, as it would be unnecessary. However, if after reading my explanation below, the authors still insist on this piece of discussion, I would require a simulation to justify, as I can only see advantages rather than disadvantages of summary-based methods compared to individual-level data methods.

The bias of CCA and MANOVA for binary phenotypes does not directly apply to summary-based methods. The methods using GWAS summary statistics for binary traits are not analyzing the binary phenotypes themselves but rather the underlying liability. This is because a logistic regression was used in the GWAS analysis, not an ordinary linear regression. This not only applies to multi-trait analysis methods but also genetic correlation analysis methods such as LDSC. This is actually an advantage of summary-based methods over individual-level data methods. Technically, a summary-based method DOES NOT KNOW the distribution of the phenotype, as we DON'T KNOW (by looking at the GWAS summary statistics) whether the betas are linear regression slopes or logistic regression log odds ratios. The underlying liability has a logistic distribution, which is nicely smooth and symmetric, thus fine for many standard

analyses.

8. Later in the Discussion, the authors were talking about how slow it is to run the "formal" test of pleiotropy, then stated, "Future development of more computationally efficient methods that use individual-level data, rather than summary statistics, would greatly facilitate the detection of pleiotropy." Is this really what was intended to be said? Isn't the current "Pleio" method using individual-level data? Shouldn't a summary-level method be much faster? I believe this is certainly doable. By looking at e.g. <https://github.com/xiashen/MultiABEL/>, it seems rather straightforward to perform summary-level multi-trait analysis on different combinations of phenotypes, though software as such has not implemented the exact sequential procedure as "Pleio" introduced.

Thank you for your feedback. We have added this citation in the discussion for the summary-level methods part (Line 343-346). Aside from the bias in the estimates and issues with model assumptions (e.g., they usually require very large N), another drawback of summary-based methods is the requirement of an external LD reference panel which might not necessarily comport with the LD distribution of the considered dataset and can pose issues in the presence of admixture.